# Notch/Her12 signalling modulates, motile/immotile cilia ratio downstream of *Foxj1a* in zebrafish left-right organizer

Barbara Tavares[1†], Raquel Jacinto[1†], Pedro Sampaio[1], Sara Pestana[1], Andreia Pinto[2], Andreia Vaz[1], Mónica Roxo-Rosa[1], Rui Gardner[3], Telma Lopes[3], Britta Schilling[4], Ian Henry[4], Leonor Saúde[5], Susana Santos Lopes[1*]

[1]CEDOC, Chronic Diseases Research Centre, NOVA Medical School - Faculdade de Ciências Médicas, Universidade Nova de Lisboa, Lisboa, Portugal; [2]Laboratório de Histologia e Patologia Comparada, Instituto de Medicina Molecular, Centro Académico de Medicina de Lisboa, Lisboa, Portugal; [3]Instituto Gulbenkian de Ciência, Oeiras, Portugal; [4]MPI of Molecular Cell Biology and Genetics, Dresden, Germany; [5]Instituto de Medicina Molecular e Instituto de Histologia e Biologia do Desenvolvimento, Faculdade de Medicina da Universidade de Lisboa, Lisboa, Portugal

**Abstract** Foxj1a is necessary and sufficient to specify motile cilia. Using transcriptional studies and slow-scan two-photon live imaging capable of identifying the number of motile and immotile cilia, we now established that the final number of motile cilia depends on Notch signalling (NS). We found that despite all left-right organizer (LRO) cells express *foxj1a* and the ciliary axonemes of these cells have dynein arms, some cilia remain immotile. We identified that this decision is taken early in development in the Kupffer's Vesicle (KV) precursors the readout being *her12* transcription. We demonstrate that overexpression of either *her12* or Notch intracellular domain (NICD) increases the number of immotile cilia at the expense of motile cilia, and leads to an accumulation of immotile cilia at the anterior half of the KV. This disrupts the normal fluid flow intensity and pattern, with consequent impact on *dand5* expression pattern and left-right (L-R) axis establishment.
DOI: https://doi.org/10.7554/eLife.25165.001

**\*For correspondence:** susana. lopes@fcm.unl.pt

[†]These authors contributed equally to this work

**Competing interests:** The authors declare that no competing interests exist.

## Introduction

Embryonic motile cilia play an essential role in body laterality patterning by generating a directional fluid flow inside the vertebrate left-right organizer (LRO) (*Nonaka et al., 1998*; *Takeda et al., 1999*; *Nonaka et al., 2002*). The mouse LRO (or node) is lined with monociliated cells, presenting motile and immotile cilia (*McGrath et al., 2003*). While motile cilia in the pit region generate a directional fluid flow (the nodal flow), the immotile cilia, present mostly in the perinodal crown cells, are thought to be able to sense it (*McGrath et al., 2003*). This sensing mechanism is still unknown and continues to deserve tentative updates in the field both using the mouse model (*Delling et al., 2016*) and the zebrafish model (*Ferreira et al., 2017*). Whether it involves mechano- or chemosensation, or both, is still not demonstrated but it is known to lead to the activation of the channel Polycystin-2 (PKD2) (*Yoshiba et al., 2012*). This channel was elegantly shown to be necessary for the asymmetric expression of *dand5* (DAN domain family, member 5) on the right side of the mouse node (*Yoshiba et al., 2012*) and consequently allowing the propagation of Nodal into the left Lateral plate mesoderm (LPM) (*Marques et al., 2004*). This signal is amplified via a self-enhanced lateral-inhibition system (SELI) (*Nakamura et al., 2006*) at the left LPM, which consists in the activation of the genetic

cascade Nodal-Pitx2-Lefty2 and ends with the correct formation and asymmetric positioning of the visceral organs (*Nonaka et al., 2002*).

In zebrafish, the left-right (L-R) axis establishment starts in a fluid-filled organ designated Kupffer's vesicle (KV) (*Essner et al., 2005*; *Kramer-Zucker et al., 2005*). Functionally, this organ is the homologue of other vertebrate LROs like the mouse node (*Nonaka et al., 2002*) and the gastrocoel roof plate in *Xenopus* (*Schweickert et al., 2007*). The KV originates from a cluster of cells, the dorsal forerunner cells (DFCs), which migrate in the forefront of the shield during gastrulation (*Cooper and D'Amico, 1996*). At the end of gastrulation, the DFCs form an ellipsoid fluid filled vesicle. While KV lumen inflates each cell extends one cilium towards the lumen (*Amack et al., 2007*; *Oteíza et al., 2008*). As in the mouse node, the KV cilia also produce a directional fluid flow that leads to an asymmetric *dand5* gene expression (*Lopes et al., 2010*; *Sampaio et al., 2014*). Our previous work determined that KV cilia can also be divided into two populations according to whether they are functionally motile or immotile (*Sampaio et al., 2014*). We also showed that the DeltaD zebrafish mutant ($dld^{-/-}$) for Notch signalling (NS) presents an increase in the number of motile cilia in the LRO (*Sampaio et al., 2014*), suggesting that NS may modulate the number of motile cilia in this organ. Equivalently, Boskovski *et al.* reported that GALNT11, an *N*-acetylgalactosamine type *O*-glycosylation enzyme needed to activate NS, regulates the ratio between motile and immotile cilia in the *Xenopus'* LRO, where less NS also increased the number of motile cilia (*Boskovski et al., 2013*). The authors showed that changing the ratio between motile and immotile cilia caused downstream defects in L-R patterning of the laterality marker *pitx2c* (*paired-like homeodomain 2*), and in the correct heart looping of *Xenopus* embryos (*Boskovski et al., 2013*).

The transcription factor Forkhead box J1a (Foxj1a) has been established as the motile cilia master regulator in the KV cells (*Stubbs et al., 2008*; *Yu et al., 2008*). Without it cilia do not form, altering the expression of L-R markers and randomizing organ *situs* (*Tian et al., 2009*). Its transcription initiates during gastrulation in the DFCs, and Foxj1a is responsible for the transcriptional activation of several motility genes, such as *dnah7* (*Choksi et al., 2014*) and *dnah9* (axonemal heavy chain dyneins that mediate the movement of cilia by hydrolysing ATP) (*Yu et al., 2008*; *Choksi et al., 2014*). This suggests that in wild type (WT) embryos, where motile and immotile cilia are present in neighbouring cells (*Sampaio et al., 2014*), Foxj1a function may be antagonized by other factors, explaining why cilia remain immotile in some cells.

In order to understand the mechanisms behind the choice of motile *versus* immotile cilia, we manipulated NS and Foxj1a levels and evaluated their impact in the ratio of motile and immotile cilia in the zebrafish LRO. We concluded that, independently from variations in *foxj1a* mRNA levels, all cilia seem to acquire a motile ultrastructure. However, NS modulates the final number of functionally moving cilia early in the KV precursors, via a mechanism that involves the activity of Her12 (hairy-related 12), a transcription repressor so far only involved in somitogenesis (*Shankaran et al., 2007*). We then experimentally showed that Her12 mediated motile/immotile cilia ratio imbalance impacts not only on fluid flow intensity but also on its pattern. This supports our previous data showing that for the KV to develop a robust fluid flow, capable of promoting normal organ *situs*, it requires a minimum of 30 motile cilia and an anterior-dorsal cluster of motile cilia (*Sampaio et al., 2014*; *Smith et al., 2014*). Overall, the evidence presented demonstrates the importance of regulating the motile/immotile cilia ratio in the generation of a robust and functional fluid flow in the zebrafish LRO.

## Results

### Cilia in the KV become motile as development progresses

As the KV starts to form and inflate, cilia begin to emerge and some start to beat, generating a fluid flow (*Essner et al., 2005*; *Kramer-Zucker et al., 2005*), while others remain static (*Sampaio et al., 2014*). How this dichotomy evolves over time is unknown. We thus questioned whether the percentage of immotile cilia at the 8-somites stage (ss) was the same since the beginning of the LRO formation. To better understand this phenomenon, we injected WT zebrafish embryos at the 1 cell stage with a titrated, non-toxic concentration of *arl13b-GFP mRNA*, and performed live imaging in a two-photon microscope. By scanning the whole KV at a low speed (0.16 frames per second, fps) in a slow scanning mode by using a high pixel dwell time (22.4 microseconds) (*Video 1*; *Figure 1A*) it was

possible to accurately distinguish the motile cilia (*Figure 1B*) from the immotile cilia (*Figure 1C*), quantify them, and track them throughout the KV development. An higher pixel dwell time allows to better identify motile from immotile cilia, thus explaining some differences in quantification of immotile cilia in comparison with another recent work (*Ferreira et al., 2017*), We then imaged the same embryos from 3 to 8 ss (5 embryos; 294 cilia); we identified every motile and immotile cilium at each time point using the 3D stacks acquired, and then tracked them throughout time in 3D projections (*Video 2*).

The results showed that at 3 ss, the vast majority of cilia were immotile (*Figure 1D*) and cilia distribution and KV cell shape were homogeneous (*Figure 1F and G*). Between 3 and 5 ss, we observed the highest rate of increase in the percentage of motile cilia (*Figure 1D, F–K*), which then tended to stabilize at 6–8 ss (*Figure 1D, L–O*). This is accompanied with a change in KV cell shape and anterior-dorsal cluster formation as previously reported (*Wang et al., 2012*). While tracking cilia, we observed two main different types of cilia behaviour (*Figure 1E*; 4 embryos; 231 cilia). We found that 79 ± 7% of cilia transitioned from immotile at 3 ss to motile at 8 ss (*Figure 1E*), while 16 ± 8% of cilia never became motile during the time-window of the assay (*Figure 1E*). We reasoned that if immotile cilia are always immotile from the beginning to the end of the KV lifetime then something should be determining the fate of such cilia very early in development.

In agreement with the increase in number of motile cilia from 3 to 8 ss, we also observed changes in the pattern of the KV fluid flow throughout development (*Figure 2*). As we had previously reported (*Sampaio et al., 2014*) we can calculate the speed and direction of the KV fluid flow by tracking native particles present in the KV by bright field microscopy (*Figure 2—videos 1–3*). Therefore, at 3–4 ss we observed a complete absence of directional fluid flow (*Figure 2A and C*; 5 embryos; 258 tracks), with the native particles presenting what seemed to be Brownian motion ($2.32 \pm 0.95$ µm s$^{-1}$; *Figure 2—video 1*; *Figure 2B*). This observation is in agreement with the low number of beating cilia found at 3 ss (6 ± 5%; *Figure 1D*). Again, here our results contrast with those from *Ferreira et al. (2017)*, which are built on numerical predictions of flow supported by a much lower number of immotile cilia. So, flow forces were calculated in different ways in the two studies, while we measured flow velocity by following native particles in live embryos averaged over a number of embryos, *Ferreira et al. (2017)* mapped cilia position and tilting in live embryos and then predicted the flow forces based on numerical simulations. As development progressed and the number of beating cilia increased (5 ss – 63 ± 23%; 6 ss – 76 ± 8%; *Figure 1D*), we began to observe a weak directional fluid flow at 5–6 ss ($6.61 \pm 3.26$ µm s$^{-1}$; *Figure 2G*), which at this time was still homogeneous throughout the KV (*Figure 2—video 2*; *Figure 2E – G*; 5 embryos; 309 tracks). Finally, at 7–8 ss, when the number of beating cilia began to stabilize (8 ss – 83 ± 6%; *Figure 1D*), the directional fluid flow acquired its characteristic velocity ($9.81 \pm 5.36$ µm s$^{-1}$) and heterogeneous pattern (*Figure 2—video 3*) (*Sampaio et al., 2014*), presenting a faster flow at the anterior-left quadrant of the KV (*Figures 2I, K* and *3* embryos, 128 tracks). While tracking native particles we could also account for cilia motility and thus by an independent method we confirmed our previous quantifications on the ratio between motile vs immotile cilia. This method was based on a high frame rate acquisitions (500 fps) using a high-speed video camera.. Again, we observed an increase in the percentage of motile cilia throughout development, ranging from 27% at 3–4 ss (4 embryos; 48 cilia; *Figure 2D*), 70% at 5–6 ss (4 embryos; 63 cilia; *Figure 2H*) to 84% at 7–8 ss (4 embryos; 89 cilia; *Figure 2L*). We also determined the cilia beat frequency (CBF) at these three different stages and we found that

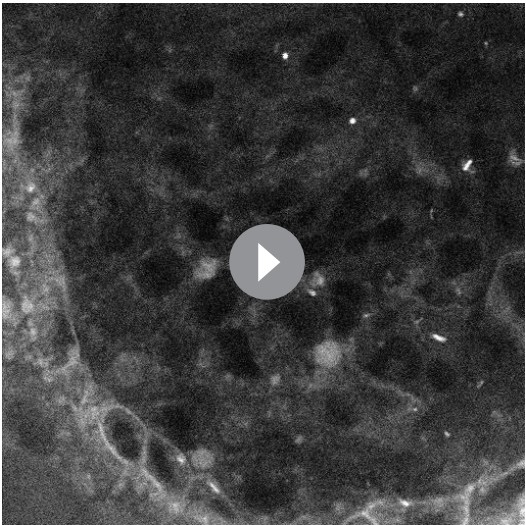

**Video 1.** Scan of Wild Type KV showing motile and immotile cilia. Embryo was injected with 400 pg Arl13b-GFP at 1 cell stage and imaged as described in *Figure 1*. Anterior is to the top and Left is to left.
DOI: https://doi.org/10.7554/eLife.25165.004

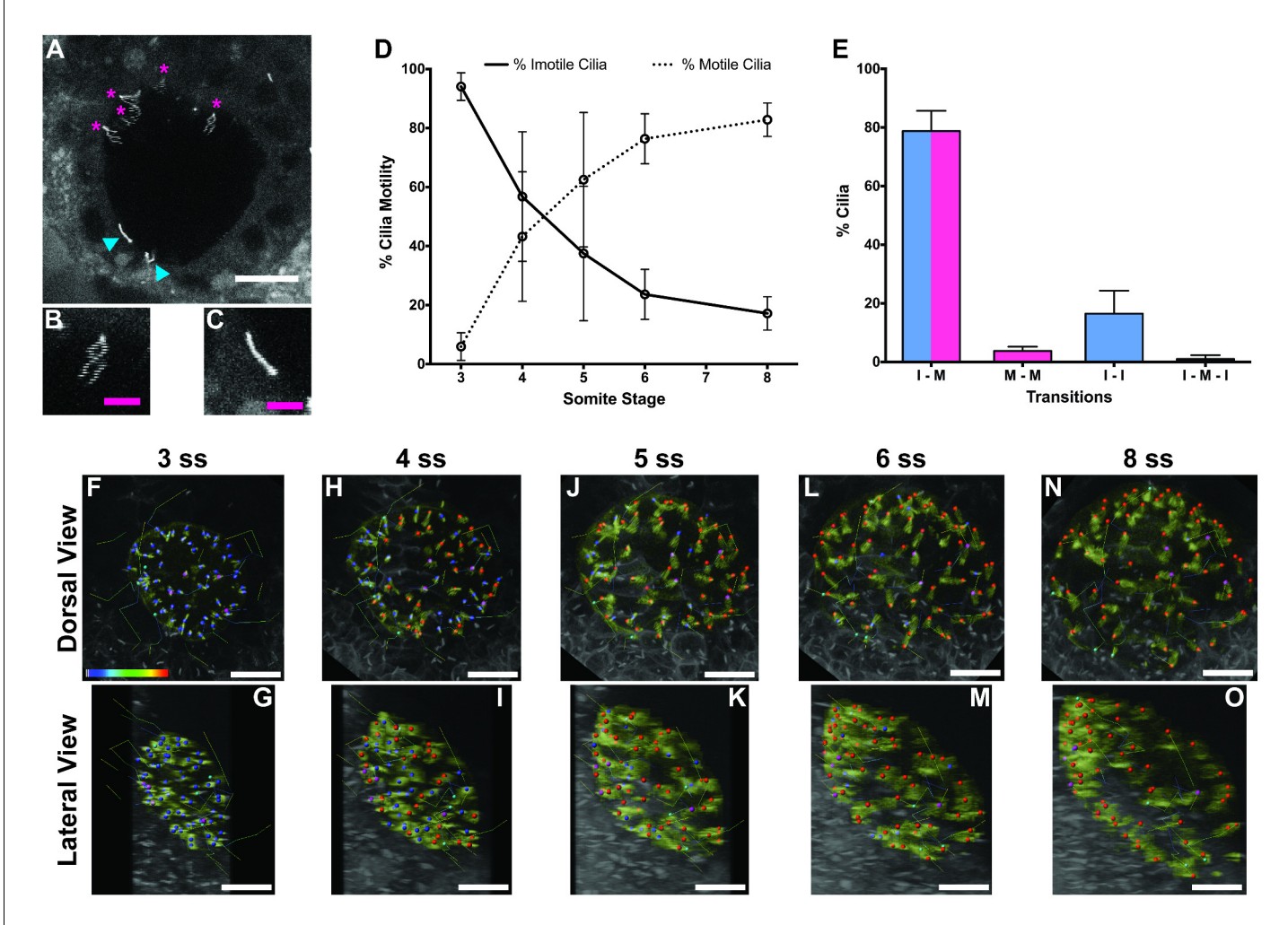

**Figure 1.** Immotile cilia are specified early in development. (**A**) Still from **Video 1** representing a wild-type embryo injected with 400 pg Arl13b-GFP at 1 cell stage. Anterior is up and Left is left. Blue arrowheads indicate – Immotile cilia; Magenta stars indicate Motile Cilia. Scale bar represents 20 μm. (**B–C**) Details of a Motile (**B**) and Immotile Cilia (**C**). Scale bar (magenta) represents 5 μm. (**D**) Changes in the % of Motile and Immotile Cilia in KV during zebrafish development from 3 to 8 somites stage ($n_e$ = 5, $n_c$ = 294). (**E**) Types of cilia motility behaviours found during the Time Lapse experiment (from 3 to 8 somites stage). I – M is 'Immotile to Motile', M – M is 'Always Motile', I – I is 'Always Immotile', and I – M – I is 'Immotile to Motile to Immotile' ($n_e$ = 4, $n_c$ = 231). (**F–O**) Stills from **Video 2**, the time-lapse video obtained from a Control embryo injected with 400 pg Arl13b-GFP at 1 cell stage. Dorsal view – Anterior is to the top and Left is to left (**F, H, J, L, N**). Lateral view – Anterior is to the top and Dorsal is to left (**G, I, K, M, O**). Motile cilia (red), Immotile Cilia (blue), cilia that were always motile (from 3 to 8 ss – purple), cilia that remain always immotile (from 3 to 8 ss – cyan). In all images, scale bar represents 20 μm. $n_e$ – number of embryos and $n_c$ – number of cilia.

DOI: https://doi.org/10.7554/eLife.25165.002

The following source data is available for figure 1:

**Source data 1.** Contains data about the number of motile and immotile cilia from single embryos along development from 3 ss to 8 ss (n = 4).

DOI: https://doi.org/10.7554/eLife.25165.003

cilia beat faster as development progressed (**Figure 2D** – L, right panels). The CBFs calculated were 33 ± 4 Hz (4 embryos; 8 cilia), 37 ± 5 Hz (4 embryos; 29 cilia), and 40 ± 4 Hz (4 embryos; 34 cilia) for the 3–4 ss, 5–6 ss and 7–8 ss, respectively. The ciliary frequency at 7–8 ss was significantly higher than at 3–4 ss. Overall the values found for the percentage of motile cilia together with the CBF may account for the differences found in the velocity of the fluid flow throughout development.

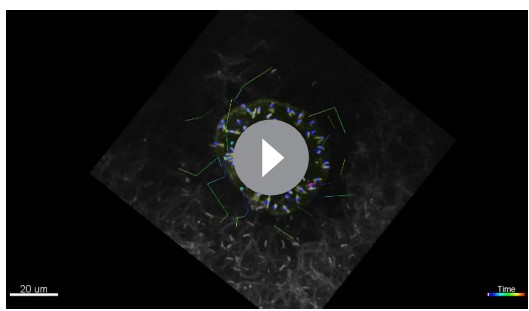

**Video 2.** Time lapse from 3 to 8 somites stage with respective cilia trackings. Tracking of the cilia was performed with Imaris software. Anterior is to the top and Left is to left.

DOI: https://doi.org/10.7554/eLife.25165.005

## Transcriptomic studies support a boost of cilia motility gene expression in deltaD mutants

We have previously reported that *DeltaD* (*dld*$^{-/-}$) mutant embryos have more cilia moving in the KV than the WT (*Sampaio et al., 2014*), indicating that Notch signalling (NS) could be responsible for the decision between motility and immotility of cilia in the KV. To understand why *dld*$^{-/-}$ had more motile cilia than control embryos we performed a comparative transcriptomic analysis of DFCs from WT and *dld*$^{-/-}$ mutant zebrafish embryos (*Figure 3A–D*). After curation, we obtained a list of 706 genes with a linear fold change (FC) in transcription higher than 2 (Table S1a in *Supplementary file 1*).

Our intention was to determine which genes in *dld*$^{-/-}$ mutants were implicated specifically in cilia motility. Therefore, we decided to use the Cildb$_{v2}$ reference base (*Arnaiz et al., 2009*) and Genevenn algorithm (*Pirooznia et al., 2007*) to determine the hits in our list that had orthologues already associated with cilia motility in other model organisms. We chose *Caenorhabditis elegans*, a primary cilia model; and *Chlamydomonas reinhardtii*, a motile cilia/flagella model (Table S1b in *Supplementary file 1*). The intersection analysis between these models permitted us to distinguish the ciliary genes specific for motility in our list. Because *foxj1a* was up-regulated (FC = 1.62, p value = 0.029) in the microarray, we also compared our gene list with the list of Foxj1a-induced genes published by *Choksi et al. (2014)*. The result of these intersections showed that 18% (129 genes) of the genes in our original list were related to ciliogenesis and of those, 67% (86 genes) were specifically related to motility (Table S1b in *Supplementary file 1*). Motility specific column). Among these were the known motility axonemal genes *dnah7* and *dnah10* (Dynein, Axonemal, Heavy Chain 7 and 10), *rsph3* and *rsph9* (Radial Spoke Head 3 and 9) together with *dld*, *foxj1a* and *rfx2*. The axonemal genes were up-regulated in *dld*$^{-/-}$ mutants and were therefore good candidates to explain the increased number of motile cilia. To validate some of the motility-associated genes obtained in the microarray, we focused on the following genes: the ciliogenesis master regulator, *rfx2* (*regulator master x 2*) (*Swoboda et al., 2000*); the motile cilia master regulator *foxj1a*, the notch ligand, *dld*; and two motile cilia axonemal components, *dnah7* and *rsph3*. We performed quantitative PCR (qPCR) using mRNA extracted from fluorescently activated cell sorted DFCs. Of these 5 tested genes, only 3 (*foxj1a*, *dld*, and *dnah7*) showed consistent changes in transcription regulation between the two methods (*Figure 3E and F*), the microarray and qPCR analyses, also confirmed by in situ hybridization in bud stage and 8 somite stage (*Figure 3—figure supplement 1*).

## Motile cilia fate decision is regulated by Notch signalling independently of Foxj1a

In order to check if Foxj1a, the master regulator of motility, was expressed in all KV cells or KV precursor cells, we decided to look at a zebrafish line that expresses GFP under a *foxj1a* minimal promoter (Tg:*foxj1a:GFP*) (*Caron et al., 2012*) and determined whether all WT KV cells were GFP positive. Our reasoning was that if we found that Foxj1a was absent in around 16–20% of WT DFCs, this alone could account for the lack of motility observed at 8 ss (*Figure 1D*). We fixed embryos at 8 ss and by immunofluorescence with an antibody against GFP and another against acetylated alpha-tubulin (*Figure 4—figure supplement 1C–E*; *Video 3*), we confirmed that the number of *foxj1a*-positive cells and the number of cilia present in each KV were concordant (*Figure 4—figure supplement 1F*; p=0.9479, 7 embryos, 514 cells, 513 cilia). This experiment showed that *foxj1a:GFP* was present in all KV cells despite the fact that some cells have immotile cilia throughout development. So, we then questioned how this *foxj1a* promoter related to the actual mRNA expression. We confirmed that, as reported before by *Yu et al. (2008)* *foxj1a* mRNA is strongly expressed in the DFCs at bud stage (*Figure 4—figure supplement 1A*) (*Yu et al., 2008*). However, this early expression is

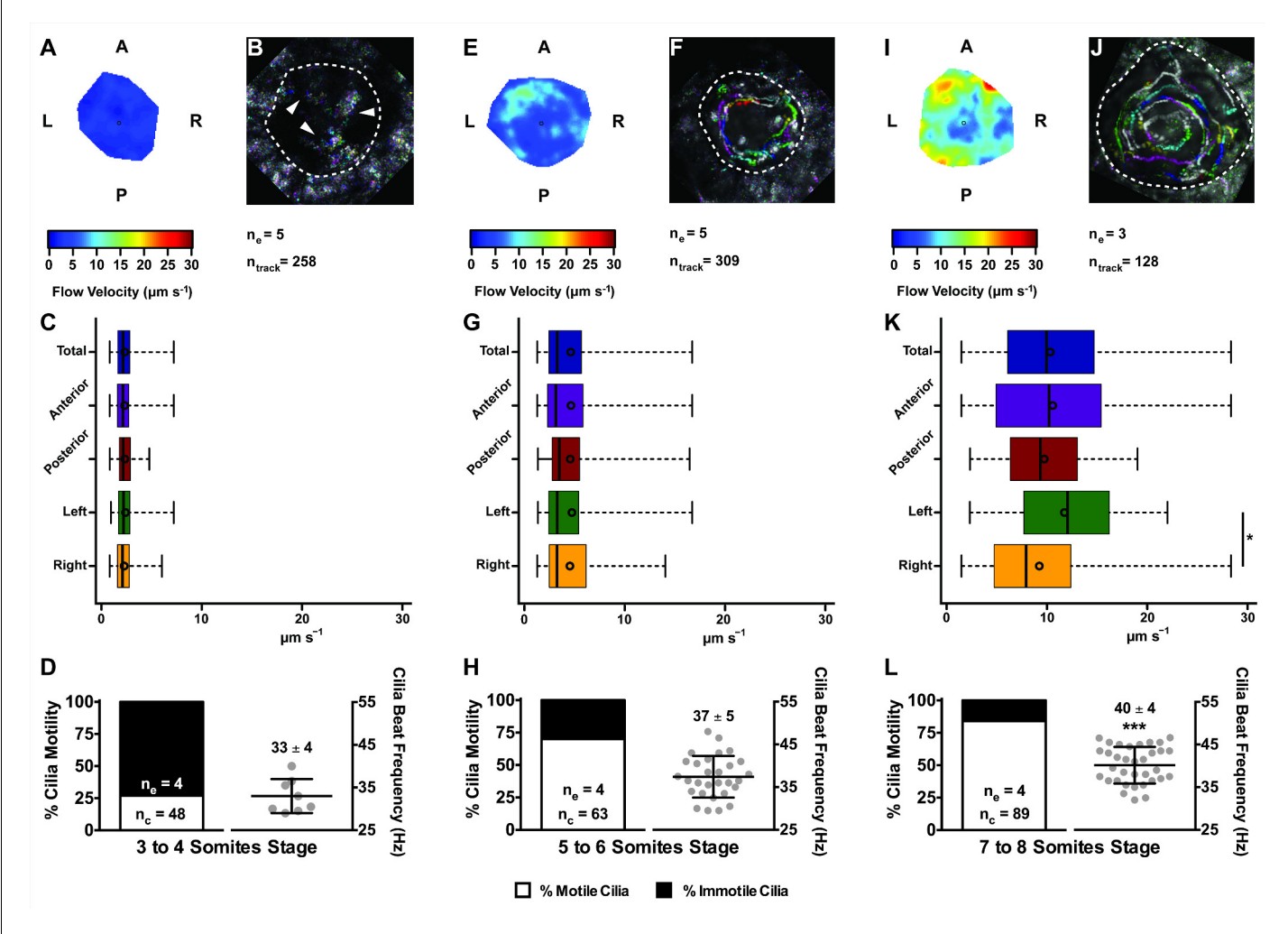

**Figure 2.** Changes in KV flow and CBF through development observed in uninjected WT embryos. Data was pooled from embryos at 3–4 (A–D), 5–6 (E–H), and 7–8 (I–L) somites stage. (A, E, I) Heat maps of flow speed showing detailed regions within the KV for pooled embryos in each experimental condition: 3–4 ss (A), 5–6 ss (E), and 7–8 ss (I). The pseudo-colour scale represents flow speed in µm s$^{-1}$, where red represents high speed versus low speed in blue. (B, F, J) Representative KV particle flow map for a WT embryo at 3–4 ss (B), 5–6 ss (F), and 7–8 ss (J). Each second is represented by a different colour. The particle Brownian motion in (B) is marked with white arrowheads. Anterior is to the top and Left is to left. (C, G, K) Box plots for instantaneous flow speed measured at different locations of the KVs, based on the same data set used to generate the heat maps, in each experimental condition: 3–4 ss (C), 5–6 ss (G), and 7–8 ss (K). Box plots display the median with a vertical line, and the whiskers represent the minimum and maximum values observed. Means are represented as small circles. *p<0.05, Wilcoxon test. (D, H, L) display the %motile and %immotile cilia found in the KV midplane (left panels) and the CBF measured in the motile cilia (right panels) of WT embryos in each experimental condition: 3–4 ss (D), 5–6 ss (H), and 7–8 ss (L). Values for CBF are Mean ±SD, ***p<0.001, ANOVA with Bonferroni's multiple comparisons test. $n_e$ – number of embryos; $n_{tracks}$ – number of tracks followed; $n_c$ – number of cilia.
DOI: https://doi.org/10.7554/eLife.25165.006

Source data 1. Contains data from the native particles tracked to generate the flow maps on *Figure 2*.
DOI: https://doi.org/10.7554/eLife.25165.010

**Figure 2—video 1.** Wild-type fluid flow at 3 ss. Movie from 1 WT, non-injected embryo. At this development stage, the native particles only present Brownian motion. Anterior is to the top and Left is to left. 30 frames per second.
DOI: https://doi.org/10.7554/eLife.25165.007

**Figure 2—video 2.** Wild-type fluid flow at 5 ss. Movie from 1 WT, non-injected embryo. At this development stage the KV presents a homogeneous directional fluid flow. Anterior is to the top and Left is to left. 30 frames per second.
DOI: https://doi.org/10.7554/eLife.25165.008

**Figure 2—video 3.** Wild-type fluid flow at 8 ss. Movie from 1 WT, non-injected embryo. At this development stage, the directional fluid flow is no longer homogeneous, presenting higher speeds at the anterior-left part of the KV. Anterior is to the top and Left is to left. 30 frames per second.
*Figure 2 continued on next page*

*Figure 2 continued*

DOI: https://doi.org/10.7554/eLife.25165.009

not yet seen with the *foxj1a*:GFP minimum promoter described by *Caron et al. (2012)* (*Figure 4—figure supplement 1*) (*Caron et al., 2012*). On the other hand, at 8 ss the mRNA expression of *foxj1a* is no longer observed in the KV, while it is detectable in the pronephros and neural tube, but the reporter shows GFP in all KV cells (*Figure 4—figure supplement 1B and C–E*, respectively). So, we must conclude that this promoter is not faithfully representing the *foxj1a* gene expression along time. However, by comparing the foxj1a *in situs* with another reporter line for DFCs, the sox17:GFP reporter, we showed that the *foxj1a* mRNA staining closely matches the sox17:GFP labelling (*Figure 4A–C*). Altogether these experiments suggest that *foxj1a* is expressed in all DFCs and that the foxj1a:GFP observed at 8 ss is due to a delayed reporter and/or GFP perdurance.

Moreover, the information that *foxj1a* was up-regulated in $dld^{-/-}$ mutant DFC cells (Table S1a in *Supplementary file 1* suggested a regulation of *foxj1a* expression by NS. Therefore, in order to understand the crosstalk between NS and *foxj1a* in regulating cilia motility, we monitored the transcription levels of *her12* (*hairy-related 12*), *dnah7* and *foxj1a* itself while manipulating NS and Foxj1a protein levels. These genes were chosen because: i) *her12* is a NS direct transcription target in the tail bud and somites (*Shankaran et al., 2007*) and in this study we found that it is highly expressed in DFCs at bud stage, and down-regulated in $dld^{-/-}$ mutants (Table S1a in *Supplementary file 1*) as previously described (*Thisse and Thisse, 2004*); and ii) *dnah7* was reported to be a transcriptional target of Foxj1a (*Choksi et al., 2014*) and an essential structural component of motile cilia, without which zebrafish KV cilia become static (*Sampaio et al., 2014*). Similar to the expression levels of *foxj1a*, we found *dnah7* significantly up-regulated in $dld^{-/-}$ mutants (*Figure 4—figure supplement 1H*) and expressed in DFCs at bud stage (Table S1a in *Supplementary file 1*). The restricted expression patterns of *her12*, *foxj1a* and *dnah7* at bud stage indicated that it was possible to use mRNA from whole embryos to determine their expression levels by qPCR (*Neugebauer et al., 2013*; *Jurisch-Yaksi et al., 2013*). As a control of this procedure, we tested *dnah9* and *rfx4*, two genes that, despite being transcribed in DFCs were not influenced by NS, according to our microarray and qPCR data (*Figure 4—figure supplement 1H*).

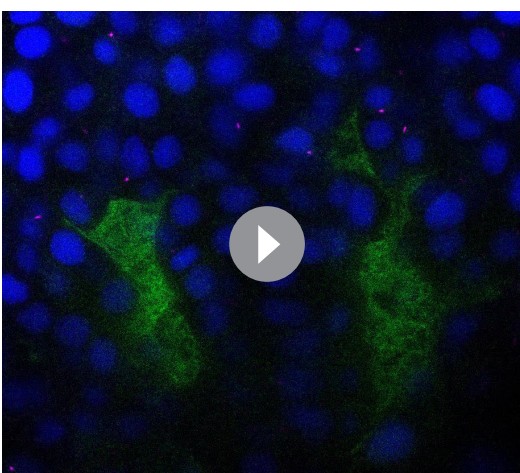

**Video 3.** Immuno-staining of a KV of an 8 ss embryo from the transgenic line Foxj1a:GFP. In blue are the nuclei stained with DAPI; in green the *foxj1a* positive KV cells; and in magenta are the KV cilia stained with antibody against acetylated α-tubulin. This experiment allowed us to determine if all monociliated KV cells expressed Foxj1a. Anterior is to the top and Left is to left.

DOI: https://doi.org/10.7554/eLife.25165.020

We then set out to manipulate the levels of NS by using the single mutants $dld^{-/-}$ and *delta like c* ($dlc^{-/-}$), and the double mutant $dlc^{-/-}$; $dld^{-/-}$. In order to increase NS we overexpressed Notch intracellular domain (NICD OE). Showing the efficiency of NS manipulations, the *her12* transcription levels behaved accordingly, being significantly lower in the $dld^{-/-}$;$dlc^{-/-}$ double mutant (*Figure 4D*), and significantly higher in the NICD OE (*Figure 4D*).

For each NS manipulation we also interfered with the levels of Foxj1a. This was assured either by knocking down the *foxj1a* expression with a morpholino against *foxj1a* translation start site (Foxj1a KD) or by overexpressing it *via* the injection of *Danio rerio foxj1a* mRNA (Foxj1a OE). As expected, the *dnah7* expression served as readout of the Foxj1a manipulation efficiency, as its expression significantly correlated both with the endogenous levels of Foxj1a (Foxj1a Not Manipulated; NM) and upon its overexpression (Foxj1a OE) (*Figure 4E*; Pearson's correlation coefficient: r = 0.8799, p value = 0.0208; and r = 0.9523, p value = 0.0034, respectively). Moreover, when compared to the WT situation, the *dnah7* mRNA levels were significantly higher upon the Foxj1a

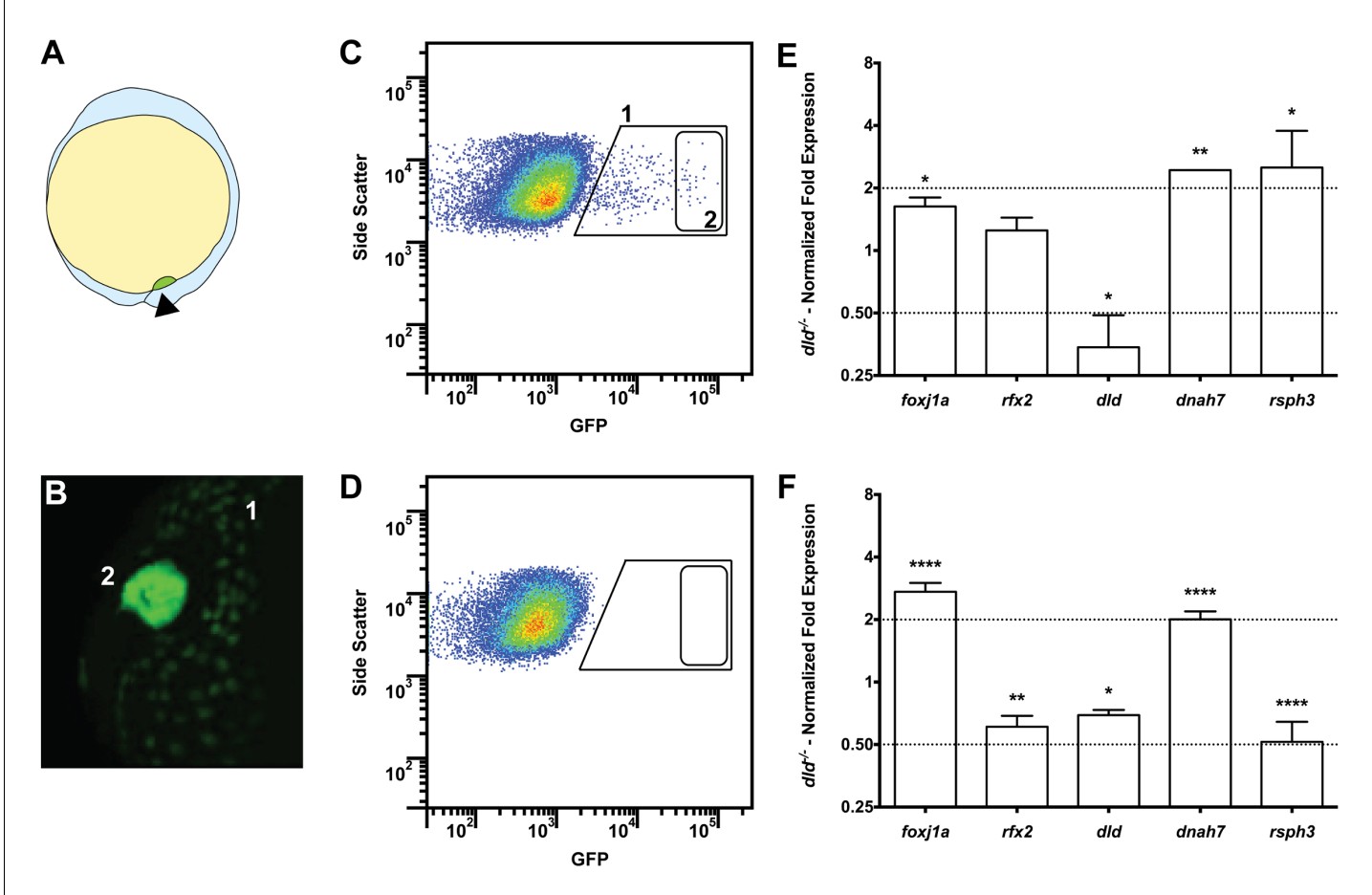

**Figure 3.** Tissue specific microarray identifies motility genes differentially expressed in $dld^{-/-}$ mutants. (**A**) Representation of a zebrafish embryo at Bud stage (10 hpf). The arrowhead shows the tail bud. In green are represented the DFCs. (**B**) At bud stage, sox17:GFP zebrafish Tg line, the DFCs strongly express GFP (2). At this same stage, endoderm cells also express GFP but at significantly lower levels (1). (**C**) FACS profiles for sox17:GFP (AB and $dld^{-/-}$) Tg lines and (**D**) WT line without GFP for assessing embryo auto-fluorescence. In (**C**) the cells contained in (1) correspond to the endoderm cells, and the cells contained in (2) correspond to the DFCs. (**E**) Normalized fold expression (log2) of several motility related genes in $dld^{-/-}$ mutant embryos as determined in the microarray. t-test; *p<0.05, **p<0.01. (**F**) qPCR validation in normalized fold expression (log2) of the microarray results in the cells selected by FACS as in (**C**). Welsh t-test or the Mann–Whitney U-test; *p<0.05, **p<0.01, ****p<0.0001.

DOI: https://doi.org/10.7554/eLife.25165.011

The following source data and figure supplement are available for figure 3:

**Source data 1.** Contains data on the relative expression levels of several genes by quantitative PCR.
DOI: https://doi.org/10.7554/eLife.25165.013
**Source data 2.** GO annotation for gene groups obtained with the R package clusterProfiler.
DOI: https://doi.org/10.7554/eLife.25165.014
**Figure supplement 1.** In situ hybridization with *dnah7* specific probe in zebrafish embryos.
DOI: https://doi.org/10.7554/eLife.25165.012

OE (*Figure 4E*). This observation is in agreement with *Choksi et al. (2014)* who previously showed that *dnah7* is induced by Foxj1a. Concordantly, the transcription levels of *dnah7* were lower in the Foxj1a KD than in the control situation (*Figure 4E*), demonstrating the efficiency of the morpholino in blocking the *foxj1a* mRNA translation. Interestingly, we observed that the changes in *dnah7* expression were dependent only on the variations in Foxj1a, irrespectively of the NS background (*Figure 4D and E*; Pearson's correlation coefficient: r = −0.1859, p value = 0.4198).

The quantification of the expression data led us to conclude that the regulation of *foxj1a* transcription by NS, uncovered in our microarray, was dependent on the NS context (*Figure 4—figure supplement 1I*). In short, the qPCR analysis showed that NS impacts indirectly on the transcription

of *dnah7* by affecting *foxj1a* transcription levels. This data confirms the results obtained in the comparative transcriptome analysis between *dld*$^{-/-}$ mutants and WT for the genes in question: *her12*, *foxj1a* and *dnah7*.

Next, to independently evaluate the impact on the combined manipulation of NS and Foxj1a levels on KV cilia motility we performed live imaging of KV cilia at 8 ss. We selected three NS conditions: *dld*$^{-/-}$;*dlc*$^{-/-}$ double mutants (low NS); WT embryos (normal NS); and embryos injected with *NICD* mRNA at 1 cell stage (high NS), while simultaneously manipulating the levels of Foxj1a by overexpressing it in the three experimental groups. We chose not to knockdown Foxj1a because this would lead to the loss of cilia altogether (*Yu et al., 2008*). We decided to use the double mutants *dld*$^{-/-}$;*dlc*$^{-/-}$ because we had previously observed laterality defects in both *dld*$^{-/-}$ and *dlc*$^{-/-}$ single mutants (*Lopes et al., 2010*) and more importantly, we wanted to avoid redundancy among the genes coding for the two Delta ligands. We filmed ciliary motility at 8 ss because by this stage we know that the anterior and left flow hotspots are established (*Figure 2I*), which we demonstrated before to be crucial for the L-R patterning (*Sampaio et al., 2014*; *Smith et al., 2014*; *Montenegro-Johnson et al., 2016*).

The live imaging analysis showed that when NS is reduced, the number of immotile cilia in the KV decreased significantly from 20 ± 5% in controls to 12 ± 4% in *dld*$^{-/-}$;*dlc*$^{-/-}$ mutants (*Figure 4F*; 24 embryos, 1047 total cilia in control; 7 embryos, 304 total cilia in *dld*$^{-/-}$;*dlc*$^{-/-}$ mutants). Conversely, when NS is increased, the number of immotile cilia increased from 20 ± 5% in controls to 31 ± 10% (*Figure 4F*; 8 embryos, 353 total cilia in NICD OE;). Importantly, NS levels interfered with the motile/immotile cilia ratio and not with the total number of cilia (24 embryos, 44 ± 12 cilia in Control versus 8 embryos, 44 ± 11 cilia in NICD OE,; p=0.9154) or with cilia length (*Figure 4—figure supplement 2A*). Regarding cilia length, we evaluated the cilia length from all treatments done in *Figure 4F*. These experiments were done using a 50 pg of arl13b-GFP so that we could observe the cilia motion live. Using the same movies we have also measured cilia length in 3D. We found no differences between any treatment (*Figure 4—figure supplement 2A*). Using the arl13b-GFP overexpression we have normalized cilia length. In this way, we could overcome the cilia length differences that Notch signaling manipulations produce and successfully uncouple cilia length from cilia motility.

So, despite the qPCR data showing a significant increase in both *dnah7* and *foxj1a* transcription levels after *foxj1a* was over-expressed (*Figure 4E* and *Figure 4—figure supplement 1I*, respectively), live imaging experiments showed that there were no changes in the actual number of motile cilia when *foxj1a* was over-expressed, irrespective of the NS background (*Figure 4F*; nine embryos, 380 total cilia in Foxj1a OE; eight embryos, 368 total cilia in *dld*$^{-/-}$;*dlc*$^{-/-}$ mutants + Foxj1 a OE; eight embryos, 347 total cilia in NICD OE +Foxj1 a OE). We conclude that the failure of Foxj1a OE to increase the number of motile cilia, and specifically the failure in rescuing the loss of motile cilia in the NICD OE embryos (*Figure 4F*) strongly suggests that the KV cells are already committed to a certain motility fate. We propose NS modulates such functional decision downstream of Foxj1a function.

Next, we investigated whether a higher level of *foxj1a* transcription, as the one found in the *dld*$^{-/-}$ mutants, was causing increased CBFs on motile cilia. However, by overexpressing Foxj1a, we observed no significant change in motile cilia CBF (WT = 37.27 ± 7.202 Hz, Foxj1a OE = 36.30 ± 6.877 Hz; p=0.4990; *Figure 4—figure supplement 1G*). This experiment also agreed with the fact that in *dld*$^{-/-}$ mutants, we did not observe an increase in CBF (*Sampaio et al., 2014*). Our results suggest that this level of Foxj1a overexpression, despite eliciting the transcription of the downstream target *dnah7* (*Figure 4—figure supplement 1I*) has no consequences in KV cilia motility, contrary to our initial assumptions.

Next, we investigated whether the Her12 transcription factor was directly involved in the regulation of the ratio between motile and immotile cilia. In order to test this hypothesis we overexpressed Her12 (Her12 OE) by injecting its mRNA at 1 cell stage and imaging the embryos at 8 ss. We observed a significant increase in the percentage of immotile cilia from 20% in controls to 27% (*Figures 4F*, 9 embryos, 305 total cilia), recapitulating the results observed in the NICD OE assay (*Figure 4F*).

Overall these observations indicate that, in parallel to the activation of the motile cilia program by *foxj1a* transcription, which should occur in all DFCs, NS decides which cilia will be stopped. Since both *foxj1a* and *her12* are expressed at bud stage in the DFCs (*Thisse and Thisse, 2004*; *Neugebauer et al., 2013*; *Jurisch-Yaksi et al., 2013*), we reason that on one hand, Foxj1a specifies

cilia capable of moving, while on the other hand, NS prevents the fulfilment of the motile cilia Foxj1a-activated program. This could be achieved either by structurally changing cilia (e.g. preventing the assembly of dynein arms), or by somehow inhibiting the motility of structurally motile cilia (e.g. switching off the dynein motors). To tackle these different scenarios in the absence of appropriate zebrafish antibodies, we cloned the coding sequence of *dnal1*, fused it to the mCherry fluorescent tag and injected this construct in *arl13b-GFP* transgenic embryos that enable live imaging of motile and immotile cilia. As Dnal1 is a light chain outer dynein arm axonemal dynein motor required for cilia movement (*Horváth et al., 2005*; *Mazor et al., 2011*), we reasoned that if only motile cilia expressed this construct then we could favour the hypothesis that motile cilia were likely to be structurally different. On the other hand if immotile KV cilia also expressed the Dnal1 construct then we could predict that all KV cilia might be structurally similar. In this last scenario, perhaps some other factor capable of switching off the dynein motors ATPase activity could be occurring.

Our results showed that Dnal1 was present in both motile and immotile KV cilia (*Figure 4G* – R; six embryos; compare K, N, Q with L, O, R). The quantification for the injection of the construct *dnal1*-mCherry was done in a sample of 56 cilia in a total of 4 embryos. In the sample of cilia positive for *dnal1*-mCherry, we saw 72% of motile cilia and 28% of immotile cilia. This ratio is not statistically different from the one for the controls used in *Figure 4F* (p=0.17, Fisher test) meaning that the *dnal1*-mCherry construct did not affect the cilia motility status and that both motile and immotile cilia can express *dnal1*-mCherry. Importantly, primary cilia from the tail region around the KV were all negative for *dnal1*-mCherry, which shows that in a mild overexpression scenario, this construct cannot enter a primary cilium (*Figure 4J,M,P*). Overall this experiment suggests that most KV cilia have dynein arms, i.e., may have the necessary machinery to move.

In order to have a definitive answer as to whether all KV cilia structurally belong to the sub-type of motile cilia, we performed transmitted electron microscopy by sampling the full KV of 3 embryos. We used 10 ss embryos and sectioned their KVs with 5–7 micron intervals (*Figure 4—figure supplement 4S*). We imaged a total of 101 cilia, providing an average of 34 cilia per KV, which according to *Figure 4—figure supplement 2B*, represents on average a random coverage of 77% of the cilia population of each KV. Our findings demonstrated that all cilia imaged had an ultra-structure characteristic of motile cilia. Some cilia had a clearly visible central pair while others showed an unfocused central pair and all cilia had visible dynein arms (*Figure 4U* - W; *Table 1*). So, we conclude that it is very likely that all KV cilia are equipped with motility apparatus.

## her12 localization in DFCs agrees with immotile cilia distribution at later stages

From the microarray and qPCR validations we discovered that *her12*, a homologue of the mammalian *hes5* that belongs to a family of transcription repressors (reviewed in (*Kageyama et al., 2007*)), was a likely candidate to mediate the cilia immotility regulation. Furthermore, we showed that overexpressing *her12* induced a significant increase in immotile cilia, and thus recapitulated the NICD overexpression phenotype. Next we asked how many cells in the cluster of DFCs were expressing *her12* at bud stage, because we reasoned that if Her12 was mediating the establishment of immotile cilia, then the number of immotile cilia should be similar to the number of *her12* positive cells. For this assay we used the transgenic line *sox17:GFP*, which allowed us to identify the DFCs while detecting *her12* expression pattern by fluorescent in situ hybridization (*Figure 5—videos 1* and *2*; *Figure 5A–F*).

Importantly, we showed that in the control situation 29 ± 8% of the DFCs were *her12* positive (*Figure 5M*; 9 embryos) which represent 10% more than the percentage of immotile cilia detected at 8 ss in the motility assay (*Figure 4F*). We also checked how was *her12* in NICD OE embryos (*Figure 5—videos 3* and *4*; *Figure 5G–L*). We found a significant increase of 14% in the percentage of *her12* positive DFCs (*Figure 5M*; from 30 to 44 ± 11%; 9 embryos). This represents a 14% more than the percentage of immotile cilia detected at 8 ss for this treatment (*Figure 4F*). This differences between *her12* positive cells and immotile cilia percentages could be explained by the existence of a fine-tuning mechanism (*Basch et al., 2016*).

Additionally, we noticed that these cells were preferentially located at the posterior half of the DFC 3-dimensional cluster (*Figure 5N*; 40 ± 5% and 60 ± 5% for the anterior and the posterior halves, respectively; 9 embryos). When NICD is over-expressed, regarding cell distribution, we

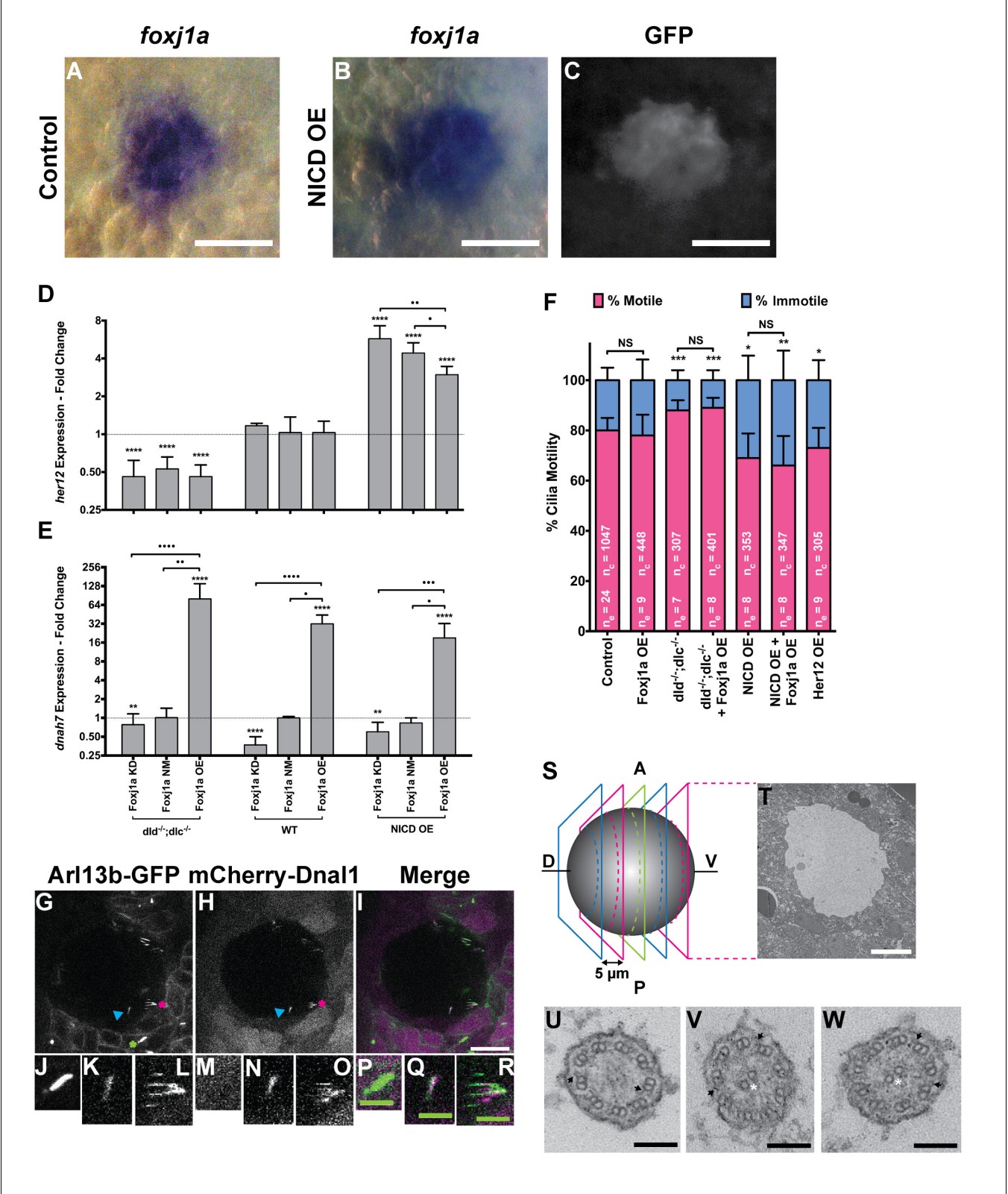

**Figure 4.** *foxj1a* is expressed in dorsal forerunner cells structurally specifying all cilia as motile cilia. (A-C) In situ hybridization with *foxj1a* at bud stage in the dorsal forerunner cells (DFCs), in a control representative embryo (A) and upon NICD overexpression (B). (C) Performing immune-staining with an antibody anti-GFP in the same embryos revealed co-localization with sox17:GFP, a marker for DFCs at bud stage. (D–E) Fold change (log2) in expression levels of *her12* (D), and *dnah7* (E) in whole embryos with different Notch Signalling and Foxj1a manipulations at bud stage. *dld⁻/⁻;dlc⁻/⁻* –
*Figure 4 continued on next page*

**Figure 4 continued**

deltaD and deltaC double mutant; WT – Wild Type, non-injected controls; NICD OE – overexpression of Notch Intracellular Domain by injecting *NICD* mRNA; Foxj1a KD – knock-down of Foxj1a by Morpholino injection; Foxj1a NM – Foxj1a non-manipulation; and Foxj1a OE – overexpression of Foxj1a by injecting *foxj1a* mRNA. Statistical significance tested with Mann-Whitney U-test (\*\*p<0.01 and \*\*\*\*p<0.0001). Kruscal-Wallis one-way analysis of variance with Dunn's correction for multiple comparisons was used to determine significant differences between different Foxj1a treatments in the same NS assay (\*p<0.05, \*\*p<0.01, \*\*\*p<0.001, and \*\*\*\*p<0.0001). (F) Changes in the % of Immotile and Motile Cilia after manipulation of NS and/or of Foxj1a levels, and imaged by Multiphoton fluorescence microscopy at 0.16 frames per second. Unpaired Welch t-test (Control vs Foxj1a OE; Control vs Her12 OE) and one-way ANOVA with Bonferroni's correction for multiple comparisons (Control vs NICD OE vs NICD OE +Foxj1 a OE; Control vs dld$^{-/-}$;dlc$^{-/-}$ vs dld$^{-/-}$;dlc$^{-/-}$ + Foxj1 a OE). \*p<0.05; \*\*p<0.01; \*\*\*p<0.001. NS stands for non significant. (G–I) Live KV from Arl13b-GFP *Tg* zebrafish embryo at 8 ss, over-expressing mCherry-Dnal1 (M–O). Of note are the positive GFP and mCherry signals present in both Immotile (K, N, Q) and Motile KV cilia (L, O, R). Primary cilium showed no mCherry signal (J, M, P). In the sample of cilia positive for *dnal1*-mCherry, we scored 72% motile cilia and 28% immotile cilia (n = 56 cilia in a total of 4 embryos). Scale bars represent 20 µm (white) and 5 µm (green). Blue arrow – immotile cilia; magenta asterisk – motile cilia; green asterisk – primary cilia. Ss -somite stage. (S–W) Transmission electron microscopy (TEM) micrographs of the kupffer's vesicle from 10 ss wild type zebrafish embryos. (S) Schematics of the sampling methodology sectioning every 5–7 microns to recover full transverse sections such as the one shown in (T). (U-W) Examples of the two types of cilia ultrastructure observed, (U) without visible central pair but showing visible outer and inner dynein arms (arrows) or with visible central pair and dynein arms (U, V).

DOI: https://doi.org/10.7554/eLife.25165.015

The following source data and figure supplements are available for figure 4:

**Source data 1.** Contains data on *foxj1a* gene expression by in situ hybridization and by qPCR.
DOI: https://doi.org/10.7554/eLife.25165.018
**Source data 2.** Relative expression levels of *dnah7, her12, dnah9, rfx4* and *foxj1a* by qPCR for different Notch signalling manipulations.
DOI: https://doi.org/10.7554/eLife.25165.019
**Figure supplement 1.** *foxj1a* expression analysis and loss and gain of function assays.
DOI: https://doi.org/10.7554/eLife.25165.016
**Figure supplement 2.** Arl13b-GFP enables live imaging of cilia motility and normalizes cilia length.
DOI: https://doi.org/10.7554/eLife.25165.017

noticed that *her12* positive cells became homogeneously distributed in both the anterior and posterior halves of the DFC cluster in contrast to the control situation (*Figure 5N*; 9 embryos).

## Posterior to anterior cell transitions induce a bias in the distribution of immotile cilia upon NICD overexpression

It is well established that KV maturation involves changes in cell shape to form an anterior-dorsal cluster (*Roxo-Rosa et al., 2015*) crucial for the anterior-left fluid flow hotspots (*Sampaio et al., 2014*; *Smith et al., 2014*; *Wang et al., 2012*; *Montenegro-Johnson et al., 2016*). However, in light of the bias of *her12* positive DFCs at the posterior part of the DFC cluster in control embryos (at bud stage *Figure 5C,D,F,N*), this led us to test a new hypothesis. Knowing that at 8 ss there is no anterior-posterior bias in the position of immotile cilia (*Sampaio et al., 2014*) we postulated that in a WT embryo with 3 ss, more cells bearing immotile cilia would be present in the posterior part of the KV, judging from the expression pattern of *her12*. We thus predict that some of those cells will be pushed to the anterior part of the KV, as previously reported (*Wang et al., 2012*) and also confirmed here, leading to a homogeneous final distribution of immotile cilia (*Figure 5O*; immotile cilia in blue) at 8 ss. On the other hand, when NICD was overexpressed, we found that the posterior bias of *her12* positive DFCs was lost (*Figure 5N*), which should lead to a homogenous distribution of immotile cilia in the KV at 3 ss, prior to cell shape changes. Therefore, upon KV maturation, we predict an accumulation of immotile cilia in the anterior half (*Figure 5P*). To test this hypothesis, we looked again to the KVs scanned at 8 ss, and used them to create 3D reconstructions and to map the position of each immotile cilia for all treatments. Using the principles of the Cartesian referential

**Table 1.** Transmitted electron microscopy sampling of cilia from the Kupffer's vesicle.

| # KV | Total number of cilia observed | Central-pair | Dynein arms | Microns covered |
|---|---|---|---|---|
| 1 | 37 | 35 | 37 | 65 |
| 2 | 40 | 39 | 40 | 70 |
| 3 | 24 | 20 | 24 | 58 |

DOI: https://doi.org/10.7554/eLife.25165.021

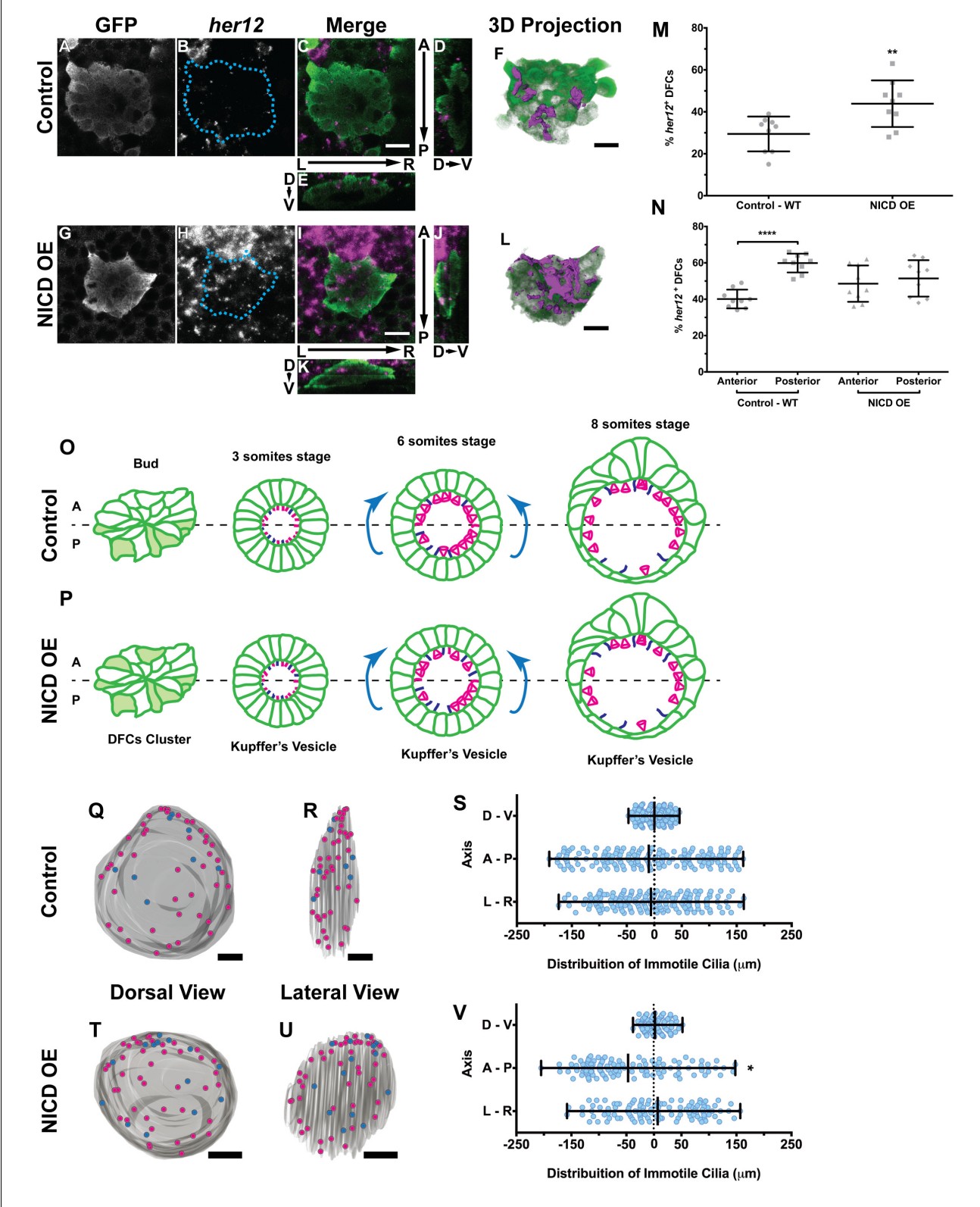

**Figure 5.** Expression of *her12* agrees with future immotile cilia *distribution*. In situ hybridization with *her12* specific probe in sox17:GFP transgenic embryos at bud stage in Control (**A–E**) and NICD over-expression embryos (**G–K**). An antibody anti-GFP was simultaneously used in order to highlight the DFCs (**A, G**). *her12* expression visualized with the Fast Red (Roche) fluorescent precipitate (**B, H**). In all images, scale bar represents 20 µm. Anterior is to the top and Left is to left. (**D–E, J–K**) Orthogonal projections emphasising *her12* expression in the DFC cluster. (**D, J**) Anterior is to the top and

*Figure 5 continued on next page*

*Figure 5 continued*

Dorsal is to left. (E, K) Dorsal is to the top and Left is to left. (M) % *her12* positive cells in the DFC cluster in Control (•) and NICD over-expression assay (■). (N) % *her12* present in the Anterior or in the Posterior halves of the DFC cluster in Control (•, ■) and NICD over-expression (p, ◆). Unpaired t-test with Welsh's correction; \*\*p<0.01 and \*\*\*\*p<0.0001. (O–P) Models describing how the cellular reorganization that transforms a DFC cluster into a fully mature KV impacts on the position of the immotile cilia in Control (O) and NICD OE (P) embryos. (Q, R, T, U) 3D projections of representative KVs, where the positions of motile (magenta dots) and immotile cilia (blue dots) are shown for Control (Q–R) and NICD OE (T–U). (Q, T) Dorsal view – Anterior is to the top and Left is to left. (R, U) Lateral view – Anterior is to the top and Dorsal is to left. In all images, scale bar represents 20 µm. (S, V) Distribution of the position of the immotile cilia along the three axes: D – V (Dorsal – Ventral); A – P (Anterior – Posterior); L – R (Left – Right), in Control (S) (17 embryos; 159 cilia) and NICD OE (V) (8 embryos; 107 cilia). Distance from centre represents the distance from the origin of the Cartesian referential (placed at the KV's centre). \*p<0.05, Fisher's Exact Test.

DOI: https://doi.org/10.7554/eLife.25165.022

**Source data 1.** Contains data on *her12* positive DFC number and its anterior posterior location within the DFC cluster.
DOI: https://doi.org/10.7554/eLife.25165.025
**Source data 2.** Provides data on the coordinates of immotile cilia denoting posterior to anterior transitions.
DOI: https://doi.org/10.7554/eLife.25165.026
**Figure supplement 1.** Immotile cilia are homogeneously distributed in the KV.
DOI: https://doi.org/10.7554/eLife.25165.023
**Figure supplement 2.** Positions of immotile cilia in the anterior-posterior axis change through development.
DOI: https://doi.org/10.7554/eLife.25165.024
**Figure 5—video 1.** In situ hybridization + immuno staining with *her12* RNA probe and antibody anti-GFP of a cluster of DFCs at bud stage from a *sox17:GFP* embryo.
DOI: https://doi.org/10.7554/eLife.25165.027
**Figure 5—video 2.** 3D reconstruction of the WT *her12* expression in the DFCs cluster at bud stage.
DOI: https://doi.org/10.7554/eLife.25165.028
**Figure 5—video 3.** In situ hybridization + immuno staining with *her12* RNA probe and antibody anti-GFP of a cluster of DFCs at bud stage from a *sox17:GFP* embryo injected with 100 pg of *NICD* mRNA at 1 cell stage.
DOI: https://doi.org/10.7554/eLife.25165.029
**Figure 5—video 4.** 3D reconstruction of *her12* expression in the DFCs cluster in an embryo over-expressing NICD at bud stage.
DOI: https://doi.org/10.7554/eLife.25165.030
**Figure 5—video 5.** 3D reconstruction of the WT DLD localization around the DFCs cluster at bud stage.
DOI: https://doi.org/10.7554/eLife.25165.031

and, establishing the centre of the KV as the origin (0, 0, 0) of the three coordinate axes, it was possible to attribute Cartesian coordinates ($x$, $y$, $z$) to each immotile cilium. Using this information, we were able to plot the distribution of the immotile cilia along the anterior-posterior, the dorsal–ventral, and the left–right axes (*Figure 5Q–V* and *Figure 5—figure supplement 1A–F*). Our results showed that the position of immotile cilia was homogeneous across all 3 axes in the control (*Figure 5Q–S*; 17 embryos; 159 cilia), when *foxj1a* is overexpressed (*Figure 5—figure supplement 1A–C*; 9 embryos; 89 cilia), and in the double mutant $dld^{-/-};dlc^{-/-}$ (*Figure 5—figure supplement 1D–F*; 7 embryos; 38 cilia). Only when NICD was overexpressed, was there a significant accumulation of immotile cilia at the anterior part of the KV (*Figure 5T–V*; 8 embryos; 107 cilia), thus supporting our hypothesis. To quantify which transitions in cell position were occurring (anterior to posterior, left to right, dorsal to ventral, or vice-versa) we revisited our time-lapse results and tracked the immotile cilia through development for the control situation. Overall, our results showed that in WT embryos, posterior to anterior transitions are the most common (*Figure 5—figure supplement 2A* – B, four embryos, nine cilia transitions, 19 immotile cilia tracked), further confirming our model. Here we show one such immotile cilium transitioning from the posterior part of the KV (at three ss), to an anterior position (at eight ss). We found a total of 4 immotile cilia transitioning from a posterior position to a more anterior position in as many embryos. If we assume an average KV with 44 ± 12 cilia (*Figure 4—figure supplement 2B*), 20% of these cilia will be immotile, giving an approximate number of 8 immotile cilia. Since at bud stage, 60% of the *her12* positive cells are located at the posterior part of the DFCs cluster, five immotile cilia (60% of total immotile cilia) start at a more posterior position, while 3 (40% of total immotile cilia) will be positioned more anteriorly. Therefore, one immotile cilia transition from posterior to anterior would suffice in order to evenly distribute immotile cilia at eight ss. This analysis is concordant with the observed number of transitions.

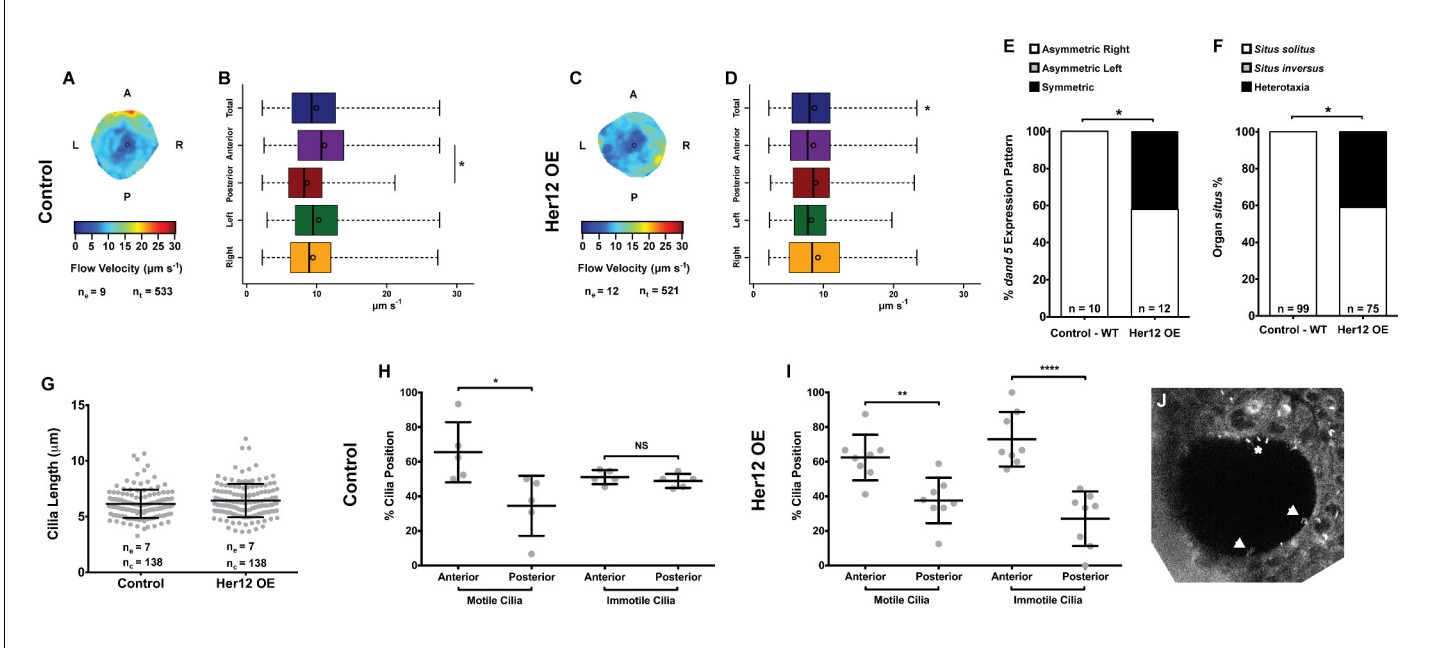

**Figure 6.** Overexpression of *her12* impacts on fluid flow dynamics, *dand5* expression pattern and organ *situs*. Fluid flow pattern and intensity found in Control (**A–B**) and in Her12 OE (**C–D**) embryos. (**A, C**) Heat maps of flow speed showing detailed regions within the KV for pooled embryos in Control (**A**) and Her12 OE (**C**). The pseudo-colour scale represents flow speed in $\mu m\ s^{-1}$, where red represents high speed versus low speed in blue. $n_e$ – number of embryos; $n_{tracks}$ – number of tracks followed. (**B, D**) Box plots for instantaneous flow speed measured at different locations of the KVs, based on the same data set used to generate the heat maps, in Control (**B**) and Her12 OE (**D**). Box plots display the median with a vertical line, and the whiskers represent the minimum and maximum values observed. Means are represented as small circles. *$p<0.05$, Wilcoxon test. (**E**) Percentages of *dand5* expression pattern at 8 ss, as determined by in situ hybridization in the same embryos used to study the fluid flow, in Control and Her12 OE. (**F**) Organ *situs* determined by observing the heart position at 30hpf, and the liver and pancreas positions at 53 hpf, by in situ hybridization with a probe for *foxa3*, in Control and Her12 OE. *Situs solitus* stands for left heart and liver; *Situs inversus* stands for right heart and liver; Heterotaxia stands for any other possible combination for the heart and liver position. *$p<0.05$, Fisher's Exact Test. (**G**) KV cilia length from 8 somite stage embryos in Control (7 embryos, 138 cilia) and Her12 OE (7 embryos, 138 cilia). The cilia were measured in 3D in fixed samples. (**H–I**) Immotile and motile cilia distribution along the Anterior – Posterior axis in Control (5 embryos) (**H**) and Her12 OE (9 embryos) (**I**) 8 somite stage embryos. Unpaired t-test with Welsh's correction; *$p<0.05$ **$p<0.01$ and ****$p<0.0001$. (**J**) Depicts a KV slice where motile (white arrows) and immotile cilia can be observed (white asterisk).
DOI: https://doi.org/10.7554/eLife.25165.032

The following source data is available for figure 6:

**Source data 1.** Provides data on flow speed and CBF upon *her12* overexpression.
DOI: https://doi.org/10.7554/eLife.25165.033

## Notch signalling occurs between DFCs and surrounding cells

Finally, the accumulation of *her12* positive cells in the posterior half of the DFC cluster of control embryos led us to question how NS was occurring, i.e. which cells were DeltaD positive. Using a DeltaD specific antibody, we looked for the expression of DeltaD at bud stage in the vicinity of DFCs (*Figure 5—video 2*; *Figure 5—figure supplement 2C–G*). This assay showed that DeltaD protein is mainly present in the cells surrounding posteriorly, laterally and dorsally to the cluster of DFCs (*Figure 5—video 5*; *Figure 5—figure supplement 2E,G and F*, respectively). We thus conclude that NS occurs between the DeltaD positive cells, forming a crescent posteriorly to the DFCs, depicted in our model in *Figure 5—figure supplement 2H*. *notch1b* positive DFCs (FC = 3.3 in Table S1a, *Supplementary file 1*) together with *notch1a*, *notch2* and *notch3* are also expressed in the DFCs at bud stage. Some DFCs, preferentially at the posterior boundary, will transcribe *her12* (*Figure 5—figure supplement 2H*) and we postulate that those are the ones where ciliary motility will be inhibited.

## Excess of anterior immotile cilia affects flow pattern, dand5 expression and organ situs

To determine whether the 7% increase in immotile cilia observed upon *her12* overexpression is relevant for the determination of laterality (*Figure 4F*) we measured fluid flow and scored *dand5* expression in the same embryos, this time without co-injecting *arl13b*-GFP. Furthermore, we scored the heart position by observing it in the live embryo at 30 hpf and the liver position at 53 hpf by ISH with a *foxa3* probe. Fluid flow analysis showed that *her12* overexpression leads to, not only a loss of the dorsal-anterior flow hotspot (compare *Figure 6A–B* with *Figure 6C-D*) but also a 12.5% reduction in flow velocity (*Figure 6B,D*; from 9.94 ± 4.47 in WT to 8.70 ± 4.26 μm s$^{-1}$ in Her12 OE; 9 and 12 embryos, respectively; 533 and 521 tracks, respectively). Regarding *dand5* expression pattern, it correlated with the difference in the fluid flow: in contrast to 100% of asymmetric on the right in control embryos (*Figures 6E*, 10 embryos), 42% of embryos overexpressing Her12 showed a symmetric expression pattern (*Figures 6E*, 12 embryos). In terms of organ *situs*, 41% of Her12 overexpressing embryos displayed heterotaxia (incorrect positioning of either the heart or liver; *Figures 6F*, 75 embryos) while 100% of the control embryos showed left heart and left liver (*situs solitus*; *Figures 6F*, 99 embryos). To test if the observed decrease in flow speed and pattern, as well as the change in *dand5* expression pattern and organ *situs* were not due to a change in cilia length we measured the cilia length from 7 KVs upon Her12 OE (without arl13b-GFP). The results showed that there was no significant difference between treated and control experiments (*Figure 6G*). In order to check if the Her12 OE really recapitulated the NICD effect leading to an increase in the anterior immotile cilia, we then counted the motile and immotile cilia that were present on the Anterior-Posterior halves of each KV. These results confirmed an increase in the number of immotile cilia at the anterior KV halves upon Her12 overexpression (*Figure 6H–I*). Altogether, we demonstrated that the change in location of immotile cilia and its accumulation at the anterior half of the KV had a significant effect in fluid flow dynamics, impacting on *dand5* expression pattern and consequently in organ *situs*.

## Discussion

In this manuscript we show the progressive increase of cilia motility in the KV from 3 ss to 8 ss using two independent methods: slow scanning two-photon microscopy allowing for cilia beat pattern trajectories to be revealed and fast acquisition transmitted light imaging by high speed videomicroscopy allowing for slow-motion video analysis (*Figures 1* and *2*). This increase in motility occurs as immotile cilia start acquiring motility throughout development as previously reported (*Yuan et al., 2015*). We report here an increase in the intensity of the fluid flow velocity together with a significant increase in the CBF from 3 to 8 ss (*Figure 2*). The changes in fluid flow velocity as the number of motile cilia increased are in line with our previous predictions (*Sampaio et al., 2014*), which dictated that a minimum of 30 cilia is necessary for a robust fluid flow showing anterior-left hotspots at 8 ss (*Smith et al., 2014*; *Montenegro-Johnson et al., 2016*). By performing live imaging in a two-photon microscope we could establish that the average number of cilia in a KV is 44 ± 12 (*Figure 4—figure supplement 2B*; 1047 cilia in 24 embryos), which according with our previous predictions (*Sampaio et al., 2014*)is more than sufficient to create a robust fluid flow at 8 ss, even accounting for the observed 20% of immotile cilia in WT embryos.

In order to understand why *dld*$^{-/-}$ mutants had more motile cilia in the KV we performed a tissue specific comparative transcriptomic analysis that showed differential expression of genes known to be Foxj1a-induced (*Choksi et al., 2014*) (Table S1b in *Supplementary file 1*) We also found that the *dld*$^{-/-}$ mutant presented a significant up-regulation of *foxj1a* (FC = 1.62), which indicated a possible regulation in transcription of this gene and its downstream targets by NS. To determine if that was in fact the case we observed the KV cilia in action, using fluorescent live imaging at 0.16 frames per second (fps) and a resolution of 512 × 512 pixels. Contrary to our expectations, we showed that the number of motile cilia was not affected by overexpressing *foxj1a*. Furthermore, we demonstrated that the balance between motile and immotile cilia is regulated by NS, similarly to what was reported in the *Xenopus* LRO (*Boskovski et al., 2013*). However we did not detect the same interaction between NS and *foxj1a* as suggested by *Boskovski et al. (2013)*. Interactions between NS and Foxj1a have also been suggested by *Jurisch-Yaksi et al. (2013)* who showed that decreased levels of Rer1 (retention in endoplasmic reticulum sorting receptor 1), lead to increased γ-secretase activity

and consequently increased NS, which lead to decreased *foxj1a* levels in zebrafish embryos (*Jurisch-Yaksi et al., 2013*). They showed this interaction occurs in several ciliated organs, such as neuromasts, pronephros, olfactory pits, and the sensory patch of the inner ear. In KV, Rer1 knockdown causes the decrease of *foxj1a* expression, but unlike in other organs, when NS is increased, *foxj1a* levels remain unchanged (*Jurisch-Yaksi et al., 2013*). This last observation concurs with our qPCR results that show that when NICD is over expressed, the expression of *foxj1a* is unchanged (*Figure 4—figure supplement 1I*).

It is also known that NS affects Gemc1/MCIDAS, which are required for Foxj1 expression and multiciliated cell fate determination (*Kyrousi et al., 2015*). In this context NS indirectly regulates the pattern of multicilliated cells in the epidermis of *Xenopus laevis* embryos and in the human airway epithelium, by the process of lateral inhibition (*Deblandre et al., 1999*; *Marcet et al., 2011a*). Additionally, the epithelia of the pronephric ducts from zebrafish embryos also present two types of cells, a multiciliated cell and a monociliated principal cell. Their differentiation and patterning is also determined by a NS-dependent lateral inhibition mechanism (*Ma and Jiang, 2007*; *Liu et al., 2007*). In this organ, multiciliogenesis is also inhibited by NS, with multiciliated cells expressing *rfx2* and *jagged2*, and principal cells expressing *notch3* and *her9* (hairy-related 9, a NS downstream target) (*Ma and Jiang, 2007*; *Liu et al., 2007*). In the three multicilliogenesis models discussed, it was suggested that NS inhibited multiciliogenesis by blocking the early expression of either *foxj1* (*Marcet et al., 2011a*; *Marcet et al., 2011b*) or *rfx2* (*Ma and Jiang, 2007*; *Liu et al., 2007*). There was an exclusion of expression patterns for *foxj1/rfx2* and *notch*, and it was this exclusion that triggered cell fate specification and differentiation. *Boskovski et al. (2013)* also suggest that NS functions upstream of Foxj1 and Rfx2 in determining the cilia type at the Xenopus LRO. Nevertheless, they never tested whether overexpressing *foxj1a* would rescue the immotility phenotype observed when NS was up-regulated (*Boskovski et al., 2013*). We were the first to show in the present study that *foxj1a* overexpression fails to rescue the immotile cilia number caused by the up-regulation of NS provided by NICD injection (*Figure 4F*). We propose here that Foxj1a acts early to specify all KV cilia as motile cilia and then Notch signalling triggers an immotility switch. Our strongest evidence comes from the random sampling by transmitted electron microscopy that clearly points to all KV cilia being of the motile sub-type. As far as we are aware we were the first to sample the whole KV by transmitted electron microscopy and to provide a high number of analysed cilia. Collectively, our results show that NS is acting downstream of Foxj1a to establish immotility.

Additionally, this study provides a new player involved in the motility decision: Her12, a paralogue of the mouse Hes5, known as a *bona fide* NS transcriptional target (*Shankaran et al., 2007*). In our hands *her12* transcription proved a faithful NS readout in the KV cell precursors. We thus suggest that DeltaD binds to Notch1a [as suggested by *Shankaran et al. (2007)*] to activate transcription of *her12*. Subsequently, Her12 perhaps together with other factors, restricts the number of motile cilia, most likely by inhibiting the early transcription of a yet unknown crucial motility switch. Here we determined that overexpressing *her12* resulted in a significant increase of immotile cilia in the anterior half of the KV, thereby softening the difference between anterior and posterior flow speed (*Figure 6A–D*). We demonstrate that this flow pattern/intensity disruption impacted on the expression pattern of *dand5* (making it symmetric) and on organ *situs* generating heterotaxia (*Figure 6E–F*). We would like to emphasize that the results presented here on motile/immotile cilia ratio were uncoupled from the cilia length defects that Notch signalling causes (*Figure 4—figure supplement 2A* and *Figure 6G*) (*Lopes et al., 2010*).

A recent work by *Ferreira et al. (2017)* determined a considerably smaller percentage of immotile cilia in the zebrafish LRO. This discrepancy can be explained by the different pixel dwell times used during acquisition in the two different studies. The pixel dwell time was ten times higher in our study, meaning that while we spent 22.4 microseconds per pixel, Ferreira et al. spent 2.4 microseconds. This parameter measures the time that is spent scanning on each pixel, so the longer the dwell time on a particular pixel, the more signal will be detected and the less it will be distorted. In opposition, the faster the scanning the more distorted becomes the imaged object and any cilia movement, even when caused passively by the KV flow bending immotile cilia, will produce a cilium blur that may be inaccurately scored as a motile cilium. Therefore, a longer pixel dwell time is crucial because it will produce a very sharp image of the trajectory of each cilium, reconstituting the beating pattern of a motile cilium (*Figure 1B*), or in the case of an immotile cilium it will show a very bright stiff cilium (*Figure 1C*). Additionally, the number of discarded cilia by Ferreira et al. may have biased

the ratio between motile and immotile cilia. We are aware that immotile cilia can be difficult to detect in 3D reconstructions due to cilia passive motion. In our own study to accurately identify immotile cilia we compared the 3D stacks side by side with the corresponding 3D reconstruction. For some immotile cilia this was crucial due to the smoother blur these may show in a 3D reconstruction. According to Ferreira et al. some embryos at 8 somites show 0% of immotile cilia which contrasts with the 20% we have detected by two independent methods.

Finally, the reason why immotile cilia detection method is important to be clarified is because *Ferreira et al. (2017)* refute the immotile cilia mechanosensory hypothesis based on their results. We therefore cannot agree with such conclusion. In fact, according to Ferreira et al. own theory for mechanosensation to work, we would need at least 3 immotile cilia on each LR side to discern noise from productive flow. Therefore based on our results and their theory, we can conclude that from 5 to 6 somite stage, we have enough immotile cilia per KV to be able to sense flow (13 cilia at 5 ss and 9 cilia at 8 ss, on average, which is more than 3 cilia on each side). So, overall we propose that the mechanosensing hypothesis cannot be excluded until demonstration of a chemosensing or mixed mechanism emerges.

Taken together it is possible to conclude that the balance between motile and immotile LRO cilia in zebrafish needs to be tightly regulated for the proper establishment of the L-R axis, by producing a robust, and most importantly, a heterogeneous patterned fluid flow. We would like to stress that in opposition to the mouse embryo (*Shinohara et al., 2012*) the fish LRO is very sensitive to motile cilia number and localization, most likely because of the different topographies of these two LROs. Whereas in mouse all motile cilia are on the floor of the node contributing to the effective flow, in zebrafish ventral pole cilia are antagonistic to the desired flow (*Smith et al., 2014*).

## Materials and methods

### Zebrafish lines

For the microarray and respective qPCR assays we used transgenic sox17:GFP zebrafish on WT AB background (gift from Carl-Philipp Heisenberg) and on deltaD/aei[tr233] homozygous mutant AB background. The later line was generated by crossing $dld^{-/-}$ mutants with the sox17:GFP line, growing the progeny to adulthood and then incrossing and selecting the GFP fluorescent $dld^{-/-}$ mutant fish by their somite phenotype (*Tübingen 2000 Screen Consortium et al., 2005*). For the motility assays (live imaging and qPCR) we used WT zebrafish line, and the homozygous double mutant line $dld^{-/-}$; $dlc^{-/-}$ line, both from AB background. This last line was obtained by crossing the homozygous mutant lines $dld^{-/-}$ and $dlc^{-/-}$, growing the progeny to adulthood and then incrossing and selecting the $dld^{-/-};dlc^{-/-}$ mutant fish by their $dlc^{-/-}$ somite phenotype, and genotyping for the presence of the aei[tr233] mutation. To genotype the deltaD/aei[tr233] mutation in $dld^{-/-};dlc^{-/-}$ double mutants, we isolated genomic DNA from individual adults' tails, amplified it and sequenced it. Homozygous mutant DeltaC/bea[tm98/tm98] fish and the transgenic zebrafish line *Tg(Foxj1a:GFP)* were gifts from Leonor Saúde lab. All these zebrafish lines were maintained and used as described in *Westerfield, 2000*. Embryos were kept at either 32°C (for Fluorescence Activated Cell Sorting, qPCR, and for CBFs measurements from 3 to 8 ss), or overnight (ON) at 25°C (for live imaging), in the dark and in E3 embryo medium, and were staged according to *Kimmel et al. (1995)*. The procedures performed to zebrafish were approved by the Portuguese Veterinary General Administration (DGAV - Direcção Geral de Alimentação e Veterinária).

### Fluorescence activated cell sorting (FACS)

Embryos at bud stage (10hpf) were dechorionated with pronase (2 mg ml$^{-1}$) and washed extensively in Danieu's buffer. Cells were dissociated by manual pipetting in $CO_2$ independent medium (Gibco) complemented with 0.5 mM EDTA. Cells were centrifuged at 700 g and re-suspended in 4 ml of the same medium (step performed 3 times). The cells were then re-suspended in 1 ml and filtered with a 30 µm filter (CellTrics) directly into a round bottom tube. FACS was performed with a FACSAria bench top High Speed Cell Sorter (Becton Dickinson) at 140 kPa (20 psi) and with a 70 µm nozzle, using a 488 nm laser and a 530/40 nm bandpass filter to excite and detect GFP, respectively. GFP positive cells present both in WT and in $dld^{-/-}$ mutant embryos were selected, collected into RTL

buffer from the RNeasy micro Kit (Qiagen Inc., Valencia, CA), and immediately frozen at −80°C. We collected three independent samples for WT and mutant embryos, each with around 20 000 cells.

## Microarray

### RNA isolation and quality evaluation

Total RNA was isolated with the RNeasy micro Kit following the manufacturer's instructions. RNA quantification was done by using a *nanophotometer P-class* (Implen) and integrity was confirmed using an Agilent 2100 Bioanalyzer for an Eukaryote total RNA pico assay (Agilent Technologies). Three biological replicates were produced with an equivalent number off cells and used in the microarray.

### Target preparation

2 ng total RNA were used to produce amplified single stranded (ss) cDNA with the Ovation Pico WTA System (NuGEN Technologies). 3 μg of the ss cDNA were used to produce double stranded (ds) cDNA with the WT-Ovation Exon Module (NuGEN Technologies). Target labelling was performed using the One-Color DNA Labelling Kit (Roche NimbleGen, Inc.) with 1 μg ds cDNA as input material. All procedures were performed according to the manufacturer's recommendations.

### Hybridization, washing and scanning

6 μg Cy3-labeled cDNA target was hybridized to Zebrafish Gene Expression 385K Arrays (Roche NimbleGen, Inc., Design ID 090506_Zv7_EXPR) following the instructions in the 'NimbleGen Arrays User's Guide: Gene Expression Analysis v3.2.

### Data analysis

Normalized datasets (Gene Calls: _RMA.calls files) were created during data extraction in the software NimbleScan according to the 'NimbleGen Arrays User's Guide. Statistical data analysis was performed using the GeneSpring GX software (Agilent Technologies, Version 11.0.2). An unpaired T-test without multiple testing corrections was applied to the data and differences in expression were considered significant when p<0.05. A list of sequence IDs was created with sequences that had a fold change in transcription >2, and expression values > 8 in at least 3 out of the 6 datasets. The list of Ensembl gene IDs was later updated to a more current Ensembl release and genome build (Ensembl release-80, ZV10) by remapping the original probes against the updated transcriptome. The list of genes was then curated to remove those with repetitive probes, unspecific probes with more than one gene target or genes with insufficient probes mapping to them. The resulting gene list was then analysed with clusterProfiler, an R package for comparing biological themes among gene clusters (http://bioconductor.org/packages/release/bioc/html/clusterProfiler.html [*Yu et al., 2012*]). The data was also analysed with Cildb$_{v2}$ (http://cildb.cgm.cnrs-gif.fr/, [*Arnaiz et al., 2009*]) and with Genevenn (http://genevenn.sourceforge.net/, [*Pirooznia et al., 2007*]) in order to filter gene targets associated with motility. These lists are in Table S1a and Table S1b in *Supplementary file 1*.

### Validation of microarray expression data

We performed quantitative PCR (qPCR) using cDNA from KV's cells at bud stage (selected as in the microarray experiment). Total RNA from WT and *dld*$^{-/-}$ mutant embryos was isolated as previously described and reverse-transcribed using iScript cDNA Synthesis Kit (Bio-Rad) according to the manufacturer's instruction. qPCR was performed on the CFX96 Real-Time PCR Detection System (BIO-RAD, Hercules, CA) using the SsoFast EvaGreen Supermix (BIO-RAD, Hercules, CA). Primers for amplification of *foxj1a* (forkhead box J1a), *dnah7* (Dynein, Axonemal, Heavy Chain 7), *rsph3* (Radial Spoke 3 Homolog), *rfx2* (Regulatory Factor X, 2), and *dld* (delta D) were designed with Primer-BLAST (NCBI). Two reference genes *sox17* (Sex Determining Region Y-Box 17) and *eef1al1* (eukaryotic elongation factor 1 alpha 1 like 1) were used. Primers can be found in Table S1c in *Supplementary file 1*. All reactions were performed with two biological replicates and three technical replicates. Results were evaluated with the Bio-Rad CFX Manager 2.0 software (BIO-RAD, Hercules, CA). Significant differences in the transcription level were determined using either the Welsh t-test or the Mann–

Whitney U-test (p<0.05), depending on the normality of the populations as established by the KS normality test.

## Injections of morpholino oligonucleotides and/or mRNA

Morpholino blocking translation of Foxj1a was used as previously described (*Tian et al., 2009*; *Echeverri and Oates, 2007*). *Danio rerio her12* coding sequence was cloned into a pCS2 +vector with the primers 5' TCAAGCTTCGAAATGGCACCCCACTCAGC (forward) and 5' CTGGAGACCC TGGTAGTCTAGAAGCGGC (reverse). *Danio rerio dnal1* coding sequence was cloned into a pCS2 +mCherry vector, with mCherry at the N-terminus of Dnal1, with the primers 5' TCAAGC TTCGAAATGGCAAAAGCAACAACTATTAAAGAGGC (forward) and 5' CGCTGGATCCTTAACTC TCCCCTTCAGTTTCC (reverse). *her12*, mCherry-Dnal1, Notch-intracellular domain (NICD) (*Takke and Campos-Ortega, 1999*), full-length *foxj1a* (*Lopes et al., 2010*), and Arl13b-GFP (a gift from Helena Soares) constructs were used to produced mRNA with the mMESSAGE mMACHINE kit (Ambion). The RNA was purified with the kit RNA Clean and Concentrator-5 (Zymo) and injected at one-cell stage at a concentration of 50 pg, 200 pg, 100 pg, 100 pg, and 400 pg respectively. Embryos were left to develop at 32°C or 25°C until the desired stage.

## Quantitative PCR

Total RNA was extracted from groups of zebrafish embryos at bud stage using the Qiagen RNeasy Mini Kit and reverse transcribed using both oligo(dT)$_{18}$ and random hexamer primers with the RevertAid First Strand cDNA Synthesis Kit following the manufacturers' instructions. This procedure was repeated for the 3 biological replicates. Expression was quantified using Roche SYBR Green I Master and the PCR was run in a Roche LightCycler 96 Real-Time PCR System with 3 technical replicates for each biological replicate. Results were analysed and depicted as fold-change of transcript levels in injected embryos relative to transcript levels in control embryos. *foxj1a*, *rfx4* (Regulatory factor X, 4), *dnah7*, *dnah9* (dynein, axonemal, heavy chain 9) *and her12* (Hairy-related 12) levels were normalized in relation to *eef1al1* and *rpl13a* (ribosomal protein L13a) expression. Significant differences in the transcription level were determined using either the Welsh t-test or the Mann–Whitney U-test (p<0.05), depending on the normality of the populations as established by the Shapiro-Wilks normality test. Significant differences between different Foxj1a treatments in the same NS assay were established with a Kruskal–Wallis one-way analysis of variance (p<0.05). Primer sequences summarized in Table S1c in *Supplementary file 1*. A biological replicate results from the cDNA produced from the total RNA extracted from 25 zebrafish embryos at bud stage. The technical replicates use the same cDNA (of each biological replicate) in the qPCR reactions. All valid replicates are used in the statistical analysis.

## Live imaging and time-lapse

Embryos were mounted live in 1% (w/v) low-melting agarose at either 2 or 7 somites stage, and covered with E3 medium. Live imaging was performed in a Prairie Multiphoton fluorescence microscope with Olympus 40x water immersion lens (NA 0.8) at 28°C. To assess motility, whole KVs were scanned with z sections of 0.5 μm, with an acquisition rate of 9.6 slices per minute (6.25 s per slice), which provided a pixel dwell time of 22,4 μs. Each experiment was repeated either two (WT vs Foxj1a OE; $dld^{-/-};dlc^{-/-}$ vs $dld^{-/-};dlc^{-/-}$ + Foxj1 a OE; WT vs NICD OE vs NICD OE +Foxj1 OE) or three times (WT vs Her12 OE). Each time an experiment was repeated, both control (WT) and tests were assayed in equal numbers until enough *n* was obtained. Time lapses were performed starting at the third somite stage and stacks were acquired every 30 min up until the sixth somite stage and 1 hr later at the eight somite stage. Since only two embryos could be imaged in this manner, this experiment was repeated 4 times in order to gather enough *n*.

## 3D identification of immotile and motile cilia and length measurement

Stacks were reconstructed and surfaces of the KVs were obtained by image segmentation in Imaris software (Bitplane, UK). Cilia were labelled with a red or blue dot based on whether they were motile or immotile, respectively. This was done for 4 embryos throughout time. Certain conditions were tracked through all time-points: immotile cilia that remained so from the first time point to the last, cilia that were motile from the beginning of the time-lapse (3 somites stage), cilia that started

immotile, became motile and then stopped moving again, and cilia that started immotile and became motile as development progressed. In order to determine the 3D coordinates of the immotile cilia in the KV, we used the stacks obtained at 8 ss in the different treatments. These were reconstructed as described, the cilia were labelled and the Cartesian coordinates (x, y, z) of each cilium were calculated by establishing the centre of the Cartesian referential as the centre of the KV.

3D cilia length were measured using the 'Simple Neurite Tracer' plugin (*Longair et al., 2011*). In live imaged KV's only immotile cilia were sampled for each condition from 8 ss embryos expressing arl13b-GFP (400 pg). In total we analysed: WT – 36 cilia, 11 embryos; Foxj1a OE – 18 cilia, 5 embryos; dld$^{-/-}$;dlc$^{-/-}$ – 24 cilia, 6 embryos; dld$^{-/-}$;dlc$^{-/-}$ + Foxj1a OE – 25 cilia, 7 embryos; NICD OE – 29 cilia, 8 embryos; NICD OE +Foxj1 a OE – 24 cilia, 7 embryos; Her12 OE – 18 cilia, 7 embryos. Specifically to compare WT and Her12 OE cilia length, 8 somite-stage embryos were fixed and the axonemal skeleton of the cilia was labeled with acetylated alpha tubulin. In this way we analyzed: WT - 138 cilia, 7 embryos; Her12 OE - 138 cilia, 7 embryos.

## Fluid flow and CBF measurements

We followed the methods described previously (*Sampaio et al., 2014*) for mounting and filming embryos, and calculating the fluid flow and the cilia beat frequency. We tracked native particles and calculated the respective flow velocity with an R script (*Supplementary file 2*) in WT embryos from 3 to 8 ss. Embryos were kept at 32°C until desired stage and then filmed at room temperature. We imaged ciliary movement for beating frequency analysis in embryos overexpressing Foxj1a at 8 ss and the corresponding non-injected WT siblings (embryos kept at 25°C until desired stage and then filmed at room temperature). Each fluid flow and CBF measurement experiments were repeated once.

## Transmission electron microscopy

Zebrafish embryos with 14 hpf (10 ss) were fixed for 16 hr at 4°C in 0.1 M sodium cacodylate buffer, pH 7.3, containing 2,5% gluteraldehyde (v/v) enriched with sucrose and calcium chloride. After washings with sucrose enriched buffer the fragments were post-fixed for 1 hr (on ice) in 1% (aq.) osmium tetroxide and contrasted in block in 1% (aq.) uranyl acetate for 30 min. Dehydration was made using ethanol gradient (50-70-95–100%). Samples were embedded in bottle neck beem capsules (Ted Pella) using EPON resin (Electron microscopy sciences) and hardened at 60°C for 72 hr. After polymerization the tip containing the embryo was sawed and the embryo was re-embed in flat silicon mold for better orientation (KV parallel to the section plan). Resin blocks were sectioned using an ultramicrotome UC7 (Leica microsystems), semi-thin sections (300 nm) were stained with toluidine blue for optic light microscopy, semi-thin were collected until the KV start appearing. Ultra-thin sections (80 nm), were obtained systematically, 2 grids with 2 sections every 5–8 µm until the end of the vesicle. Sections were collected into formvar coated copper slot grids (AGAR scientific), and counter-stained with uranyl acetate and lead citrate (Reynold recipe), the whole KV was screened in a Hitachi H-7650 transmission electron microscope at 100kV acceleration, cilia were tilted in order to check for ultrastructural features, pictures were taken using a XR41M mid mount AMT digital camera.

## Antibody staining, immunofluorescence, immuno-in situ hybridization, and confocal microscopy

We followed the methods described previously (*Neugebauer et al., 2009*) for immunostaining. The mixed immunofluorescence and in situ hybridization technique was adapted (*Thisse and Thisse, 2008*) as follows: on the second day, we added the antibody anti-GFP together with the antibody Anti-Digoxigenin-AP Fab Fragments (Roche) and incubated over night at 4°C in a horizontal rotator; on the third day, we added the secondary antibody anti-rabbit Alexa Fluor 488 and incubated over night at 4°C in a horizontal rotator; on the fourth day, we developed the RNA probe with Fast-Red substrate (Roche) until a red deposit was observed and the reaction was stopped with 4% PFA (in PBS) for 5 min. *her12* RNA probe was a gift from Leonor Saude. Antibodies used were anti-acetylated α-tubulin (1:300; T7451 from Sigma), anti-GFP (1:500; ab290 from Abcam), anti-DlD (1:50; zdd2 monoclonal antibody [*Itoh et al., 2003*]), anti-mouse Alexa Fluor 564, and anti-rabbit Alexa Fluor 488 (1:500; Invitrogen). Nuclei were stained with DAPI (1:500). Flat-mounted embryos were

examined with a Zeiss LSM 710 Meta confocal microscope and a Zeiss 40x water immersion C-Apochromat lens (1.2 NA). Three-colour confocal z-series images were acquired using sequential laser excitation, and analysed using Fiji software (LSM Reader) (*Schindelin et al., 2012*).

## In situ hybridization *on* whole-mount embryos

Whole-mount in situ hybridization was performed as described previously (*Thisse and Thisse, 2008*). Digoxigenin RNA probes were synthesized from DNA templates of *dnah7* (*Sampaio et al., 2014*), *dand5* (*Hashimoto et al., 2004*), *foxa3* (*Monteiro et al., 2008*) and *foxj1a* (*Yu et al., 2008*). Images of flat-mounted embryos were acquired in a Zeiss Z2 Widefield Microscope with Zeiss air EC Plan-Neofluar 5x (0.16 NA) and 10x (0.3 NA) lenses. Whole-mount in situ hybridization to detect the expression of *dand5* and *foxa3*.

## Data statistical analysis

Statistical analysis was performed with Prism 6 and R (Wilcoxon test and Fisher's Exact Test). Data populations were tested for normality with the Shapiro-Wilks or the KS normality tests and the different statistical tests were used accordingly. These are specified in the materials and methods section and on the Figures' legend. The values presented are Means ± SD, unless stated otherwise.

# Acknowledgements

The authors wish to thank Julian Lewis for the DeltaD antibody, Carl-Philipp Heisenberg for the gift of the transgenic zebrafish line Tg(sox17:GFP), Leonor Saúde for the gift of the transgenic zebrafish line *Tg(foxj1a:GFP)*, the homozygous mutant DeltaC/bea[tm98/tm98], and the *her12 in situ* probe, and Helena Soares for the Arl13b-GFP construct. We would like to thank Idan Tuval, Domingos Henrique, Rita Teodoro and Adán Guerrero for insightful discussion of the manuscript. We also want to thank the IGC Advanced Image Facility, mainly to Gabriel Martins for advice and the use of Imaris software, and the IGC and CEDOC Fish Facilities. R.J. was supported by the Mechanisms of Disease and Regenerative Medicine (ProRegeM) PhD programme.

# Additional information

### Funding

| Funder | Grant reference number | Author |
| --- | --- | --- |
| Fundação para a Ciência e a Tecnologia | PTDC/BEX-BID/1411/2014 | Susana Santos Lopes |
| Fundação para a Ciência e a Tecnologia | FCT-ANR/BEX-BID/0153/2012 | Sara Pestana |
| Fundação para a Ciência e a Tecnologia | PTDC/SAU-OBD/103981/2008 | Andreia Vaz |
| Fundação para a Ciência e a Tecnologia | PD/BD/52420/2013 | Raquel Jacinto |
| Fundação para a Ciência e a Tecnologia | SFRH/BPD/77258/2011 | Barbara Tavares |
| Fundação para a Ciência e a Tecnologia | SFRH/BD/111611/2015 | Pedro Sampaio |
| Fundação para a Ciência e a Tecnologia | IF/00951/2012 | Susana Santos Lopes |

The funders had no role in study design, data collection and interpretation, or the on the decision to submit the work for publication.

### Author contributions

Barbara Tavares, Raquel Jacinto, Conceptualization, Data curation, Formal analysis, Investigation, Methodology, Writing—original draft, Writing—review and editing; Pedro Sampaio, Data curation, Formal analysis, Investigation, Methodology, Writing—review and editing; Sara Pestana,

Investigation, Methodology; Andreia Pinto, Investigation, Visualization, Methodology; Andreia Vaz, Formal analysis, Investigation, Methodology; Mónica Roxo-Rosa, Validation, Writing—review and editing; Rui Gardner, Data curation, Formal analysis; Telma Lopes, Britta Schilling, Data curation, Formal analysis, Methodology; Ian Henry, Data curation, Formal analysis, Validation, Investigation, Methodology; Leonor Saúde, Writing—review and editing; Susana Santos Lopes, Conceptualization, Formal analysis, Supervision, Funding acquisition, Investigation, Methodology, Project administration, Writing—review and editing

### Author ORCIDs
Raquel Jacinto, http://orcid.org/0000-0002-4029-0204
Susana Santos Lopes, http://orcid.org/0000-0002-6733-6356

### Decision letter and Author response
Decision letter https://doi.org/10.7554/eLife.25165.037
Author response https://doi.org/10.7554/eLife.25165.038

---

## Additional files

### Supplementary files
• Supplementary file 1. Microarray data. Excel file that contains Table S1a - List of 706 genes with significantly altered transcription. This list contains 706 genes with a fold change in transcription higher than 2, in the DFCs from $dld^{-/-}$ mutant zebrafish embryos. Table S1b – List of motility associated genes from the Table S1a that have been associated with cilia in the different model organisms. Analysis performed with Cildb v2. Table S1c – List of primers sequences used for genotyping $dld^{-/-}$ mutant zebrafish embryos and for qPCR validations.
DOI: https://doi.org/10.7554/eLife.25165.034

• Supplementary file 2. Contains the R script for creating and analysing the flow maps.
DOI: https://doi.org/10.7554/eLife.25165.035

• Transparent reporting form
DOI: https://doi.org/10.7554/eLife.25165.036

---

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
