## [Decision Letter]

Thank you for submitting your article "Notch/Her12 signalling sets the ratio of motile/immotile cilia independently of Foxj1a in zebrafish left-right organizer" for consideration by *eLife*. Your article has been favorably evaluated by Marianne Bronner (Senior Editor) and four reviewers, one of whom is a member of our Board of Reviewing Editors. The reviewers have opted to remain anonymous.

The reviewers have discussed the reviews with one another and the Reviewing Editor has drafted this decision to help you prepare a revised submission.

The manuscript by Taveres et al. addresses the role of Notch signaling in determining the ratio of motile/immotile cilia in Kupffer's Vesicle (KV) through its target, *her12*. The left-right organizers (LROs) in mouse, *Xenopus* and zebrafish are known to contain both motile and immotile cilia which has led to the two-cilia hypothesis, where two types of cilia are required to produce and sense flow during left-right patterning. Experiments in the *Xenopus* LRO by Boskovski (2013) put forth evidence that Notch signaling affects the ratio of motile to immotile cilia and this was proposed to occur by suppressing Foxj1 and RFX gene expression, critical regulators required for motile cilia to form. However, work, mainly by the Lopes lab, has shown that Notch signaling promotes longer cilia in zebrafish LRO, a finding seemingly at odds with those in *Xenopus*, since motile cilia tend to be longer than sensory cilia. The current manuscript attempts to resolve some of these issues by examining genes downstream of Notch signaling in the zebrafish LRO. The results presented are extensive. The overall impression is that the conclusions are not suitable for *eLife* but that upon revision the work may be.

The first two figures document when immotile and motile cilia arise developmentally as the LRO forms in Zebrafish and how this correlates with the appearance of flow. These experiments ground the subsequent analysis. The authors then present a transcriptomic analysis of Delta mutants, finding that Foxj1 and motile cilia components appear to be Notch regulated, but then the story gets complicated. First, in Figure 3 they argue that Foxj1 is not differentially regulated in the LRO, despite the differential expression in the microarray. The clear limitation with the experiment is that they use a minimal promoter driving GFP to carry out this analysis, and one wonders whether the regulation via Notch is simply not capture by this reporter. There is no attempt to show that the minimal promoter is a faithful proxy for endogenous *foxj1a* expression in the KV. Ideally Foxj1a protein would be tracked through an antibody staining, but if that is not possible then an in situ for *foxj1a* expression (similar to what was done for *her12* in Figure 4) is necessary to show that *foxj1a* expression is uniform throughout the KV. Along the same lines, from the image shown in Figure 2, there may be differing levels of *foxj1a::GFP* expression in some cells – could all cells be green, but there be significant differences in expression that could account for the reasonably subtle effects on motility that are observed?

In addition, the authors do not ask whether the expression of this reporter changes when Notch signaling is altered. They show qPCR results in Figure 4—figure supplement 1 but the results are also complicated: Foxj1a goes up in single Delta mutants but is largely unchanged in the double mutants, and NICD has no affect. Finally, the ratio between motile and immotile cilia in the LRO when Notch signaling or Foxj1 activity is manipulated barely changes in Figure 3, even when Notch ICD, or *her12*, overexpression is used. This result is not compatible with the idea that Notch/Her12 signaling has a major impact on motile cilia formation. The simplest interpretation of these results is that neither Notch nor Fox and may suggest that the rather limited input seen with Notch is unlikely to be direct. The data shown in Figure 4 also do not provide much in the way of additional support for the author's conclusion that Notch/*her12* is a major regulator of the motile/immotile decision. The number of *her12* expressing cells goes up to 70% in a NICD experiment, but the number of immotile cilia only changes from the wild type by maybe 5% on average. The authors also focus on the number of anterior versus posterior immotile cilia in NICD injected embryo but again these results document subtle differences. Some comment should be made on the relative importance of Notch/*her12* pathway compared to other regulators of ciliary motility.

Figure 3, Imaging at 0.16 fps seems to slow to quantify reliably the rotational speed of motile cilia. This might be the reason that the authors did not detect any difference in H. The reference the authors provide (Sampiao et al) images at 500fps, which is a 10 fold oversampling in relation to CBF and sufficient to resolve cilia movement. The authors should repeat the experiment using faster imaging, as, if it turns out that the CBF is reduced after NICD or Her12 overexpression, the interpretation of the data will change. It would also be helpful to know the frame rates used in the other figures. For example, what frame rates for imaging motile cilia 4Q-V?

The authors present data in Figure 6, suggesting that a subtle change in the motile/immotile cilia ratio causes heterotaxy. However, the author's own work in Development (2010) shows that Notch signaling also affects the length of cilia, which one might predict would affect ciliary flow in KV. The authors have not examined cilia length in any of the experiments reported here, making it unclear how they dissociate length versus number of motile/nonmotile cilia in terms of generating the phenotype. The authors need to address the confounding issue of effects on ciliary length versus motility/immotility.

[Editors' note: further revisions were requested prior to acceptance, as described below.]

Thank you for resubmitting your work entitled "Notch/Her12 signalling modulates motile/immotile cilia ratio downstream of Foxj1a in zebrafish left-right organizer" for further consideration at *eLife*. Your revised article has been favorably evaluated by Marianne Bronner (Senior Editor), a Reviewing Editor, and one reviewer.

The manuscript has been improved but there are some remaining issues that need to be addressed before acceptance, as outlined below:

One reviewer still has substantive concerns about this work. As you will read, most of the points can be addressed with textual changes. One point will require analysis of data that I suspect that you may have in hand. The final point, which the reviewer thinks is important, may require an additional loss of function experiment.

*Reviewer #3:*

The revised manuscript by Lopes and colleagues has improved over the previous version, although technical difficulties prevent them from addressing several critical issues. The main strength of the manuscript is that it raises some interesting ideas about the nature of immotile cilia in the KV, and how they might arise. It further fuels the controversy about whether flow is sensed by immotile sensory cilia, or by another mechanism during left-right patterning. It also raises the possibility that Notch is influencing immotile cilia differentiation, in a mechanism that is independent of Foxj1, in contrast to what has been reported in *Xenopus* by the Khoka lab. The authors have included some additional data, like the TEM analysis, which adds further to the considerable amount of data included in the manuscript. At the same time, there are several issues I think still could be addressed to make this a manuscript suitable for *eLife*.

1) The first two figures document how the ratio of motile to immotile cilia changes over time during the formation of the KV. The recent *eLife* paper from Ferreira et al. (2017), which carried out a similar analysis, reaches significantly different conclusions. I am sympathetic to the author's plight, since their work is under review, as a competing manuscript is published. Nonetheless, I feel that there needs to be a more thorough discussion of the marked discrepancy between the two papers (now only briefly mentioned in passing in the Discussion), beginning in the Results section. Otherwise, we have two *eLife* papers reaching different conclusions about the number of motile versus immotile cilia in the KV, leaving the general reader with the impression that scoring these numbers is very unreliable. This discrepancy, for example, is critical when assessing the fact that manipulating Notch signaling only causes a very small number of cilia to switch from immotile to motile, or vice versa.

2) The previous review asked the authors to validate their conclusion based on the microarray analysis that decreasing Notch signaling in the KV (which leads to a decrease in immotile cilia) increases the expression of Foxj1 and some key targets of Foxj1 required for cilia motility. Apparently this is technically difficult to do since the promoter reporter doesn't recapitulate the normal expression of Foxj1 and the in situ protocol they are using does not have the resolution they need to examine expression in the small number of cells with immotile cilium, where this expression would be predicted to change. And so in the end, whether or not Foxj1 or motile gene expression is affected by Notch in the KV remains unknown. The expression data on Her12 seems clear. However that on *foxj1* and *dnah7* is not: microarray data reports a change, but then the authors seem to want to conclude they are not changed based on low-resolution detection methods that wouldn't necessarily detect the critical changes in a small percentage of cells. In addition, the quantitative PCR does not support a consistent change as predicted by microarray results. The authors could do a much better job summarizing the expression data than the current one written in the fifth paragraph of the subsection “Motile cilia fate decision is regulated by Notch signalling independently of Foxj1a”. In addition, the two supplemental figures (Figure 3—figure supplement 1 and 2) describing G0 term analysis of the microarray data seems not to present information that is informative. I would highly recommend removing these data from the manuscript, especially since the microarray data have not been validated in a meaningful way.

3) Figure 5 and Figure 6 relate to the expression of Her12 as a Notch target gene that the authors conclude is required for setting the ratio of motile to immotile cilia. It is rather surprising that this conclusion is reached without a loss-of-function analysis, showing that Her12 normally plays a role in cilia motility and left-right patterning. This seems to be a major omission for a critical point made in the manuscript.

4) The DnaI1-GFP data in Figure 4 are potentially valuable since they support the author's suggestion that the immotile cilia in the KV are not simply sensory cilia, but have motile features. However, the analysis of Dnal1-GFP in Figure 4 cannot be published as is, and needs to be quantified by examining localization over a large number of cilia.

5) The authors previously showed that manipulating Notch in the KV affects overall cilia length and left-right patterning. In this paper, they argue that they bypass this effect by overexpressing Arl13b (which causes cilia lengthening) so that they can then assess how left-right patterning is affected by a change in the ratio of motile and immotile cilia upon alteration by Notch. This might be kosher, but how Arl13b promotes cilia lengthening is not well understood (maybe more membrane than axoneme?). By using Arl13b to correct cilia length, have they really corrected the effects of Notch on cilia length and function they previously reported? This technical limitation is significant and really needs to be discussed explicitly in a way that makes this caveat clear, so that the reader understands the limitations in the data in Figure 6.

---

## [Author Response]

The manuscript by Tavares et al. addresses the role of Notch signaling in determining the ratio of motile/immotile cilia in Kupffer's Vesicle (KV) through its target, her12. The left-right organizers (LROs) in mouse, Xenopus and zebrafish are known to contain both motile and immotile cilia which has led to the two-cilia hypothesis, where two types of cilia are required to produce and sense flow during left-right patterning. Experiments in the Xenopus LRO by Boskovski (2013) put forth evidence that Notch signaling affects the ratio of motile to immotile cilia and this was proposed to occur by suppressing Foxj1 and RFX gene expression, critical regulators required for motile cilia to form. However, work, mainly by the Lopes lab, has shown that Notch signaling promotes longer cilia in zebrafish LRO, a finding seemingly at odds with those in Xenopus, since motile cilia tend to be longer than sensory cilia. The current manuscript attempts to resolve some of these issues by examining genes downstream of Notch signaling in the zebrafish LRO. The results presented are extensive. The overall impression is that the conclusions are not suitable for eLife but that upon revision the work may be.The first two figures document when immotile and motile cilia arise developmentally as the LRO forms in Zebrafish and how this correlates with the appearance of flow. These experiments ground the subsequent analysis. The authors then present a transcriptomic analysis of Delta mutants, finding that Foxj1 and motile cilia components appear to be Notch regulated, but then the story gets complicated. First, in Figure 3 they argue that Foxj1 is not differentially regulated in the LRO, despite the differential expression in the microarray. The clear limitation with the experiment is that they use a minimal promoter driving GFP to carry out this analysis, and one wonders whether the regulation via Notch is simply not capture by this reporter. There is no attempt to show that the minimal promoter is a faithful proxy for endogenous foxj1a expression in the KV. Ideally Foxj1a protein would be tracked through an antibody staining, but if that is not possible then an in situ for foxj1a expression (similar to what was done for her12 in Figure 4) is necessary to show that foxj1a expression is uniform throughout the KV.

We agree with the reviewer that this transgenic needs further validation. So, we have now confirmed that by in situ hybridization there is no *foxj1a* expression at 8 somite-stage in the KV (Figure 4—figure supplement 1). However, all our transcriptomic analysis is done at bud and relates to the dorsal forerunner cells (DFCs) where *foxj1a* mRNA is expressed in the DFCs, as reported by several authors such as, Yu et al. Nat Genet. 2008; Tian et al. Biochem Biophys Res Commun. 2009; Lopes et al. Development. 2010. However, at bud stage the *foxj1a* minimal promoter (Caron et al. 2009) is not yet driving GFP expression in the DFC cluster.

We now show, by in situ hybridization, that *foxj1a* mRNA is present in the DFC cluster at bud stage (Figure 4 and Figure 4—figure supplement 1). So, we can now conclude that DFC cells express *foxj1a*, meaning that the *foxj1a:GFP* we reported at 8 somite-stage is likely due to a delay in the reporter line or to GFP perdurance. Nevertheless it shows that *foxj1a* was expressed in those cells. The older Figure 3 was moved to Figure 4—figure supplement 1 and replaced by DFC cluster *foxj1a* in situ hybridization and sox17:GFP staining.

We would like to highlight that we did not use the *foxj1a:GFP* line for any experiments apart from making the point that all ciliated KV cells have expressed *foxj1a* during their developmental path.

Along the same lines, from the image shown in Figure 2, there may be differing levels of foxj1a::GFP expression in some cells – could all cells be green, but there be significant differences in expression that could account for the reasonably subtle effects on motility that are observed?

That is an interesting question. We now checked this possibility by analyzing live imaging of motile and immotile cilia (labeled by arl13b-GFP) in embryos from the *foxj1a:GFP* line at 8 somite-stage. We quantified the fluorescent levels in every cell and checked if cells with motile cilia had higher levels of fluorescence than cells with immotile cilia, as suggested by the reviewer. Although we found some differences in GFP fluorescence intensity levels, we found no significant correlation between this and motility (See Author response image 1). We decided to show this data here and did not include it in the manuscript because we did not use this line for any further analysis in our study. We could still include it if the referee finds this pertinent.

**Author response image 1. respfig1:** *foxj1a::GFP* expression levels do not differ significantly between motile and immotile-cell type. Ratio of average fluorescence intensity values between immotile (total n = 18) and motile cilium cell type (total n = 22) is shown for each KV analyzed (n = 5).

In addition, the authors do not ask whether the expression of this reporter changes when Notch signaling is altered.

As suggested by the reviewer we now know that this reporter is not fully suited for this kind of experiment so we decided to proceed in another way. We looked for *foxj1a* expression by in situ hybridization at bud stage upon NICD OE. We did this experiment using a sox17:GFP reporter line that specifically labels DFCs at bud stage. The results showed that the two labeling coincide in the DFC cluster (KV precursor cells). However, we could not provide single cell resolution because the fast red development stopped working. We repeated the experiment 5 times in the last month and changed all the buffers and reagents but still it is not working. With more time we could troubleshoot what is going wrong. We decided to include the images from the NBT-BCIP purple staining and the corresponding Sox17:GFP labeling after immune-staining with anti-GFP (Figure 4). In order to do it in 3D we need the fast red staining to work. However, faced with the new results from the ultrastructural data we decided to go ahead with the resubmission since we provide evidence that all cells are likely expressing *foxj1a* in the DFCs because we only detected motile cilia in the KV.

They show qPCR results in Figure 4—figure supplement 1 but the results are also complicated: Foxj1a goes up in single Delta mutants but is largely unchanged in the double mutants, and NICD has no affect.

Indeed, the fact that neither the double Delta mutants nor NICD have impact on *foxj1a* mRNA suggests that Notch signaling has no effect on *foxj1a* expression under these manipulations. We checked this again by in situ hybridization to look for any impact on expression pattern rather than on levels of transcripts. Results show that comparing the DFC in situ hybridization staining between WT and NICD there are no apparent differences in pattern (Figure 4).

Finally, the ratio between motile and immotile cilia in the LRO when Notch signaling or Foxj1 activity is manipulated barely changes in Figure 3, even when Notch ICD, or her12, overexpression is used. This result is not compatible with the idea that Notch/Her12 signaling has a major impact on motile cilia formation. The simplest interpretation of these results is that neither Notch nor Fox and may suggest that the rather limited input seen with Notch is unlikely to be direct.

Indeed overexpressing Foxj1a does not change the ratio and that is exactly our point because we argue that only Notch signaling does that. Notch signaling manipulations make a 10% difference in the motility ratio. We concur that this may seem mild and that it may show an indirect regulation by Notch, but based on our previous work we argue that 10% difference is relevant. The justification comes from numerical simulations reported in Sampaio et al. (2014). We have concluded that a minimum number of 30 motile cilia is necessary to ensure a correct left-right in zebrafish. So, if increasing Notch signaling results in less 10% of motile cilia this will have an impact in left-right as Figure 6 demonstrates. In Figure 4—figure supplement 2, we present data on total cilia number for 24 embryos; 1047 cilia. Based on this data, on average, one KV shows 41 cilia (SD = 7.318), of which only 80% are motile, meaning on average each KV has 33 motile cilia. If we now take 10% of these motile cilia due to NICD manipulation, this number would drop to 30 motile cilia. This average number is dangerous because it is borderline the predicted minimum 30. Meaning that some embryos would have less than 30.

In addition, the localization of these immotile cilia is crucial. If they are placed in the dorsal anterior cluster they might disrupt it and abolish the anterior faster speed characteristic of all the WT embryos and greatly responsible for the correct left–right output (as demonstrated in many of our recent papers; Sampaio et al., 2014; Smith et al., 2014 and Montenegro–Johnson et al., 2016).

The data shown in Figure 4 also do not provide much in the way of additional support for the author's conclusion that Notch/her12 is a major regulator of the motile/immotile decision. The number of her12 expressing cells goes up to 70% in a NICD experiment, but the number of immotile cilia only changes from the wild type by maybe 5% on average.

The reviewer focused on the maximum and minimum values. The average value for *her12* positive cells is 40% for NICD OE and the number of immotile cilia is 30% for the same manipulation. So, we interpret these 10% discrepancy as suggestive of a fine-tuning process typical in development and agree that it also opens the possibility for other players being involved.

The authors also focus on the number of anterior versus posterior immotile cilia in NICD injected embryo but again these results document subtle differences. Some comment should be made on the relative importance of Notch/her12 pathway compared to other regulators of ciliary motility.

Yes, the difference may be subtle as we explained above but still be relevant because of the two variables at stake:

1) number of motile cilia must be above 30

2) position of the immotile cilia should not disrupt the anterior dorsal cluster. In the flow map from Figure 6, we can appreciate that the flow at the dorsal anterior corner is weaker and that the bias in the anterior vs. posterior flow was lost (see flow plot). We now analyzed where are the immotile cilia upon *her12* OE by live imaging and scored their position in each KV half in the anterior-posterior axis. We show that there is a clear increase in the number of immotile cilia at the anterior half of the KV explaining the weaker flow speed at this position.

Figure 3, Imaging at 0.16 fps seems to slow to quantify reliably the rotational speed of motile cilia. This might be the reason that the authors did not detect any difference in H. The reference the authors provide (Sampiao et al) images at 500fps, which is a 10 fold oversampling in relation to CBF and sufficient to resolve cilia movement. The authors should repeat the experiment using faster imaging, as, if it turns out that the CBF is reduced after NICD or Her12 overexpression, the interpretation of the data will change. It would also be helpful to know the frame rates used in the other figures. For example, what frame rates for imaging motile cilia 4Q-V?

We apologize for not being clear enough in the Materials and methods section. There are two very different aims here and each had its own specific imaging requirements:

1) For distinguishing motile versus immotile cilia we used 0.16 fps in a slow scanning mode in a 2-photon microscope. Under these conditions we can unequivocally identify all immotile cilia in the whole KV in 3D (see Video 1). This slow scanning method was used to produce the data in Figure 4, Figure 5, Figure 5—figure supplement 1 and Figure 6.

2) To evaluate cilia beat frequency (CBF) shown in Figure 4—figure supplement 1, we used 500 fps provided by high-speed video-microscopy described in Sampaio et al. 2014 for the midplane of the KV. Therefore, we are sure that there are no differences between WT and foxj1a OE regarding CBF. As for NICD and *her12* OE we did not evaluate CBF because the focus of the study was on the ratio between motile vs. immotile cilia. The high-speed videomicroscopy system that we have does not allow us to reconstruct the KV in 3D and does not allow fluorescence imaging (yet).

The authors present data in Figure 6, suggesting that a subtle change in the motile/immotile cilia ratio causes heterotaxy. However, the author's own work in Development (2010) shows that Notch signaling also affects the length of cilia, which one might predict would affect ciliary flow in KV. The authors have not examined cilia length in any of the experiments reported here, making it unclear how they dissociate length versus number of motile/nonmotile cilia in terms of generating the phenotype. The authors need to address the confounding issue of effects on ciliary length versus motility/immotility.

We totally agree with the reviewer that this was a major weakness of the study. We have now addressed this point and provide extensive data on cilia length. We first evaluated the cilia length from all treatments in Figure 4. These experiments were done using a 50 pg of arl13b-GFP so that we could observe the cilia motion live. Using the same videos we have now measured cilia length. We used a 3D measuring tool named Simple Neurite Tracer a plugin for FIJI (Longair MH, Baker DA, Armstrong JD. Simple Neurite Tracer: Open Source software for reconstruction, visualization and analysis of neuronal processes. Bioinformatics 2011).

We found no differences between treatments (Figure 4—figure supplement 2). Using the arl13b-GFP overexpression we have normalized the cilia length. In this way, we could overcome the cilia length differences that Notch signaling manipulations produce and successfully uncouple cilia length from cilia motility.

Furthermore, in Figure 6 we showed that laterality defects were produced by *her12* OE without using arl13b-GFP. We suggested that defects on *dand5* expression pattern and organ situs are likely caused by incorrect speed and defective flow pattern (Figure 6). Therefore, it was important to check if *her12* overexpression (alone) could produce any cilia length phenotype. Our results showed that overexpressing *her12* at 50 pg had no impact on cilia length. This new data was included in the main Figure 6 as well as in the text.

[Editors' note: further revisions were requested prior to acceptance, as described below.]

The manuscript has been improved but there are some remaining issues that need to be addressed before acceptance, as outlined below:One reviewer still has substantive concerns about this work. As you will read, most of the points can be addressed with textual changes. One point will require analysis of data that I suspect that you may have in hand. The final point, which the reviewer thinks is important, may require an additional loss of function experiment.Reviewer #3:The revised manuscript by Lopes and colleagues has improved over the previous version, although technical difficulties prevent them from addressing several critical issues. The main strength of the manuscript is that it raises some interesting ideas about the nature of immotile cilia in the KV, and how they might arise. It further fuels the controversy about whether flow is sensed by immotile sensory cilia, or by another mechanism during left-right patterning. It also raises the possibility that Notch is influencing immotile cilia differentiation, in a mechanism that is independent of Foxj1, in contrast to what has been reported in Xenopus by the Khoka lab. The authors have included some additional data, like the TEM analysis, which adds further to the considerable amount of data included in the manuscript. At the same time, there are several issues I think still could be addressed to make this a manuscript suitable for eLife.1) The first two figures document how the ratio of motile to immotile cilia changes over time during the formation of the KV. The recent eLife paper from Ferreira et al. (2017), which carried out a similar analysis, reaches significantly different conclusions. I am sympathetic to the author's plight, since their work is under review, as a competing manuscript is published. Nonetheless, I feel that there needs to be a more thorough discussion of the marked discrepancy between the two papers (now only briefly mentioned in passing in the Discussion), beginning in the Results section. Otherwise, we have two eLife papers reaching different conclusions about the number of motile versus immotile cilia in the KV, leaving the general reader with the impression that scoring these numbers is very unreliable. This discrepancy, for example, is critical when assessing the fact that manipulating Notch signaling only causes a very small number of cilia to switch from immotile to motile, or vice versa.

Thank you for this opportunity. We agree that the manuscript from Ferreira et al. 2017 is an elegant and important study that deserves to be properly discussed in our manuscript. We have now included more discussion points along the manuscript including a reference in the Introduction, notes in the Results and Discussion sections. We think the comparison of the two studies brings novelty to the field but most importantly it should make us hold on a bit longer to the mechanosensory hypothesis.

The main difference in our studies is the identification of the number of immotile cilia present in each KV. We have identified 2 sources for this discrepancy:

1) pixel dwell times are different

2) number of discarded cilia in Ferreira et al. study

The consequence of the higher number of immotile cilia that we detected compared to Ferreira et al. is that according to their own theory, the mechanosensory hypothesis would still hold true because we detect more than 3 immotile cilia on each KV side.

So, we have inserted the following comments in the manuscript:

“By scanning the whole KV at a low speed (0.16 frames per second, fps) in a slow scanning mode by using a high pixel dwell time (22.4 microseconds) (Video 1; Figure 1) it was possible to accurately distinguish the motile cilia (Figure 1) from the immotile cilia (Figure 1), quantify them, and track them throughout the KV development. An higher pixel dwell time allows to better identify motile from immotile cilia, thus explaining some differences in quantification of immotile cilia in comparison with another recent work [Ferreira et al., 2017], We then imaged the same embryos from 3 to 8 ss (5 embryos; 294 cilia); we identified every motile and immotile cilium at each time point using the 3D stacks acquired, and then tracked them throughout time in 3D projections (Video 2).”

“Again, here our results contrast with those from Ferreira et al. [Ferreira et al., 2017], which are built on numerical predictions of flow supported by a much lower number of immotile cilia. […] This method was based on a high frame rate acquisitions (500 fps) using a high-speed video camera.”

Note (not in the text)

To experimentally validate their modulations the authors redirect the readers to data from Supatto et al. 2008. After careful screening of this referenced paper we could not find detailed data referring to particle tracks on precise developmental stages that we could use to compare with our own experimental data. More importantly, the flow experiments seem to start only after 6 somite-stage therefore there is no information on earlier stages. On the other hand, we are reporting that at 3 somite-stage we did not detect directional flow in contrast to Ferreira et al. predicted modulations. These discrepancies are important examples that emerge from using different methods.

In the Discussion section, the fifth paragraph now reads:

“A recent work by Ferreira et al. (2017) determined a considerably smaller percentage of immotile cilia in the zebrafish LRO [Ferreira et al., 2017]. […] So, overall we propose that the mechanosensing hypothesis cannot be excluded until demonstration of a chemosensing or mixed mechanism emerges.”

2) The previous review asked the authors to validate their conclusion based on the microarray analysis that decreasing Notch signaling in the KV (which leads to a decrease in immotile cilia) increases the expression of Foxj1 and some key targets of Foxj1 required for cilia motility. Apparently this is technically difficult to do since the promoter reporter doesn't recapitulate the normal expression of Foxj1 and the in situ protocol they are using does not have the resolution they need to examine expression in the small number of cells with immotile cilium, where this expression would be predicted to change. And so in the end, whether or not Foxj1 or motile gene expression is affected by Notch in the KV remains unknown.

We would like to clarify these points a bit better. The reviewer is correct. However, we do not question that there are different levels of *foxj1a* mRNA under different Notch signaling manipulations. We, in fact show microarray and qPCR data confirming these differences. What we argue is that those differences are not responsible for the change in motility ratio between motile and immotile cilia, since when we overexpress Foxj1a we still see the same ratio as before (Figure 4). These observations lead us to think that all cells have motile cilia but something dependent on Notch is stopping 20% of these cilia at 8 ss. This hypothesis is now backed-up by the EM data that we have added to the manuscript.

We also wrote regarding the *foxj1a* minimum promoter that it has a problem in time but not in space: “we must conclude that this promoter is not faithfully representing the *foxj1a* gene expression along time” and then we conclude:

“Altogether these experiments suggest that *foxj1a* is expressed in all DFCs and that the foxj1a:GFP observed at 8 ss is due to a delayed reporter and/or GFP perdurance”.

So the differences in *foxj1a* mRNA levels that we detected by microarray in *dld*^-/-^ mutants and were confirmed by qPCR are not responsible for the motility ratio phenotype. They were the clue that started this story but ended up not being the reason behind the motility phenotype. We suggest here in this manuscript that the reason is downstream of Her12.

The expression data on Her12 seems clear. However that on foxj1 and dnah7 is not: microarray data reports a change, but then the authors seem to want to conclude they are not changed based on low-resolution detection methods that wouldn't necessarily detect the critical changes in a small percentage of cells. In addition, the quantitative PCR does not support a consistent change as predicted by microarray results.

Regarding the qPCR data (Figure 4—figure supplement 1) we show that the single mutant for *dld*^-/-^ and *dlc*^-/-^ both up-regulate *foxj1a* and therefore *dnah7*, confirming the microarray results.

The authors could do a much better job summarizing the expression data than the current one written in the fifth paragraph of the subsection “Motile cilia fate decision is regulated by Notch signalling independently of Foxj1a”. In addition, the two supplemental figures (Figure 3—figure supplements 1 and 2) describing G0 term analysis of the microarray data seems not to present information that is informative. I would highly recommend removing these data from the manuscript, especially since the microarray data have not been validated in a meaningful way.

We summarized the data the best we can, as requested by the reviewer, and have removed the figures as suggested.

3) Figure 5 and Figure 6 relate to the expression of Her12 as a Notch target gene that the authors conclude is required for setting the ratio of motile to immotile cilia. It is rather surprising that this conclusion is reached without a loss-of-function analysis, showing that Her12 normally plays a role in cilia motility and left-right patterning. This seems to be a major omission for a critical point made in the manuscript.

Not addressed because reviewer withdrew the request after further clarifications.

4) The DnaI1-GFP data in Figure 4 are potentially valuable since they support the author's suggestion that the immotile cilia in the KV are not simply sensory cilia, but have motile features. However, the analysis of Dnal1-GFP in Figure 4 cannot be published as is, and needs to be quantified by examining localization over a large number of cilia.

We apologize for this mistake we should have included this quantification before.

The revised manuscript now reads:

“The quantification for the injection of the construct *dnal1*-mCherry was done in a sample of 56 cilia in a total of 4 embryos. In the sample of cilia positive for *dnal1*-mCherry, we saw 72% of motile cilia and 28% of immotile cilia. […] Overall this experiment suggests that most KV cilia have dynein arms, i.e., may have the necessary machinery to move.”

5) The authors previously showed that manipulating Notch in the KV affects overall cilia length and left-right patterning. In this paper, they argue that they bypass this effect by overexpressing Arl13b (which causes cilia lengthening) so that they can then assess how left-right patterning is affected by a change in the ratio of motile and immotile cilia upon alteration by Notch. This might be kosher, but how Arl13b promotes cilia lengthening is not well understood (maybe more membrane than axoneme?). By using Arl13b to correct cilia length, have they really corrected the effects of Notch on cilia length and function they previously reported? This technical limitation is significant and really needs to be discussed explicitly in a way that makes this caveat clear, so that the reader understands the limitations in the data in Figure 6.

Thank you for this concern we think we have now clarified this issue in the text. We have published on cilia length and Arl13b this year. The mechanism that leads to cilia length increase seems to be membrane derived and not IFT derived. However it is still not clear how the tubulin cytoskeleton follows the membrane growth. We have studied the impact of longer KV cilia on flow and found it is surprisingly rather mild (Dynamics of cilia length in left–right development. P. Pintado, P. Sampaio, B. Tavares, T. D. Montenegro-Johnson, D. J. Smith, S. S. Lopes. R. Soc. open sci. 2017 4 161102; DOI: 10.1098/rsos.161102. Published 8 March 2017).

We have inserted these comments on the text:

“Regarding cilia length of Figure 4, we evaluated the cilia length from all treatments done in Figure 4. […] In this way, we could overcome the cilia length differences that Notch signaling manipulations produce and successfully uncouple cilia length from cilia motility.”

Regarding cilia length of Figure 6:

We have not used arl13b co-injection for the cilia length measurements in Figure 6, nor to the left-right flow maps. We show that *her12* OE does not affect cilia length and thus it really uncouples the cilia length defect that we saw with NICD back in 2010 from the motility issue (Lopes et al. Development. 2010). We have explained this better in the text relating to Figure 6.